# Interpreting Categorical Distributional Reinforcement Learning: An Implicit Risk-Sensitive Regularization Effect

## Abstract

The theoretical advantages of distributional reinforcement learning (RL) over expectation-based RL remain elusive, despite its remarkable empirical performance. Starting from Categorical Distributional RL (CDRL), our work attributes the potential superiority of distributional RL to its *risk-sensitive entropy regularization*. This regularization stems from the additional return distribution information regardless of only its expectation via the return density function decomposition, a variant of the gross error model in robust statistics. Compared with maximum RL that explicitly optimizes the policy to encourage the exploration, we reveal that the resulting risk-sensitive entropy regularization of CDRL plays a different role as an augmented reward function. It implicitly optimizes policies for a risk-sensitive exploration towards true target return distributions, which helps to reduce the intrinsic uncertainty of the environment. Finally, extensive experiments verify the importance of this risk-sensitive regularization in distributional RL, as well as the mutual impacts of both explicit and implicit entropy regularization.

## 1 Introduction

The intrinsic characteristics of classical reinforcement learning (RL) algorithms, such as Q-learning (Sutton & Barto, 2018; Watkins & Dayan, 1992), are based on the expectation of discounted cumulative rewards that an agent observes while interacting with the environment. In stark contrast to the classical expectation-based RL, a new branch of algorithms called distributional RL estimates the full distribution of total returns and has demonstrated the state-of-the-art performance in a wide range of environments (Bellemare et al., 2017a; Dabney et al., 2018b;a; Yang et al., 2019; Zhou et al., 2020; Nguyen et al., 2020; Sun et al., 2022b). Meanwhile, distributional RL also inherits other benefits in risk-sensitive control (Dabney et al., 2018a), policy exploration settings (Mavrin et al., 2019; Rowland et al., 2019), robustness (Sun et al., 2023) and optimization (Sun et al., 2022a).

Despite the existence of numerous algorithmic variants of distributional RL with remarkable empirical success, we still have a poor understanding of what the effectiveness of distributional RL is stemming from, and the theoretical advantages of distributional RL over expectation-based RL are still less studied. Previous works (Lyle et al., 2019) proved that in many realizations of tabular and linear approximation settings distributional RL behaves the same as expectation-based RL under the coupling updates method, but it diverges in non-linear approximation. Both risk-neutral and risk-averse domains were investigated in offline distributional RL (Ma et al., 2021). However, there is still a gap between the theory and practice, especially in the non-linear function approximation case.

In this paper, we illuminate the behavior difference of distributional RL over expectation-based RL starting from Categorical Distributional RL (CDRL) (Bellemare et al., 2017a), the first successful distributional RL family. Within the Neural Fitted Z-Iteration framework, we decompose the distributional RL objective function into an expectation-based term and a risk-sensitive entropy regularization via the *return density function decomposition*, a variant of the gross error model in robust statistics. As such, (categorical) distributional RL can be interpreted as a risk-sensitive entropy regularized Neural Fitted Q-Iteration. More importantly, the resulting entropy regularization serves as an augmented reward in the actor-critic framework, leading to a different policy exploration strategy compared with maximum entropy (MaxEnt) RL. Instead of explicitly optimizing the entropy

of policies to encourage the exploration in MaxEnt RL, the risk-sensitive entropy regularization in CDRL implicitly optimizes policies to explore states along with the following policy-determined actions whose return distributions lag far behind the target return distributions. A theoretically principled algorithm called *Distribution-Entropy-Regularized Actor Critic* is also proposed accordingly, interpolating between expectation-based and distributional RL. Empirical results substantiate the crucial role of risk-sensitive entropy regularization effect from CDRL in the potential superiority over expectation-based RL on both Atari games and MuJoCo environments. We also reveal the mutual impacts of both implicit risk-sensitive entropy in distributional RL and explicit vanilla entropy in MaxEnt RL, providing more potential research directions in the future.

## 2 PRELIMINARY KNOWLEDGE

In classical RL, an agent interacts with an environment via a Markov decision process (MDP), a 5-tuple $(\mathcal{S}, \mathcal{A}, R, P, \gamma)$, where $\mathcal{S}$ and $\mathcal{A}$ are the state and action spaces, respectively. $P$ is the environment transition dynamics, $R$ is the reward function and $\gamma \in (0, 1)$ is the discount factor.

**Action-State Value Function vs Action-State Return Distribution.** Given a policy $\pi$, the discounted sum of future rewards is a random variable $Z^\pi(s, a) = \sum_{t=0}^{\infty} \gamma^t R(s_t, a_t)$, where $s_0 = s$, $a_0 = a$, $s_{t+1} \sim P(\cdot|s_t, a_t)$, and $a_t \sim \pi(\cdot|s_t)$. In the control setting, expectation-based RL focuses on the action-value function $Q^\pi(s, a)$, the expectation of $Z^\pi(s, a)$, i.e., $Q^\pi(s, a) = \mathbb{E}[Z^\pi(s, a)]$. Distributional RL, on the other hand, focuses on the action-state return distribution, the full distribution of $Z^\pi(s, a)$. We call its density as *action-state return density function*, which we use constantly.

**Bellman Operators vs Distributional Bellman Operators.** In the policy evaluation of classical RL, the value function is updated via the Bellman operator $\mathcal{T}^\pi Q(s, a) = \mathbb{E}[R(s, a)] + \gamma \mathbb{E}_{s' \sim p, a' \sim \pi}[Q(s', a')]$. We also define Bellman Optimality Operator $\mathcal{T}^{\text{opt}} Q(s, a) = \mathbb{E}[R(s, a)] + \gamma \max_{a'} \mathbb{E}_{s' \sim p}[Q(s', a')]$. In distributional RL, the action-state return distribution of $Z^\pi(s, a)$ is updated via the distributional Bellman operator $\mathfrak{T}^\pi$, i.e., $\mathfrak{T}^\pi Z(s, a) \overset{D}{=} R(s, a) + \gamma Z(s', a')$, where $s' \sim P(\cdot|s, a)$ and $a' \sim \pi(\cdot|s')$. The equality implies random variables of both sides are equal in distribution. We use this random-variable definition of $\mathfrak{T}^\pi$, which is appealing and easily understood due to its concise form, although its return-distribution definition is more mathematically rigorous (Rowland et al., 2018; Bellemare et al., 2022).

**Categorical Distributional RL (CDRL).** CDRL (Bellemare et al., 2017a) can be viewed as the first successful distributional RL algorithm family that approximates the return distribution $\eta$ by a discrete categorical distribution $\hat{\eta} = \sum_{i=1}^{N} p_i \delta_{z_i}$, where $\{z_i\}_{i=1}^{N}$ is a set of fixed supports and $\{p_i\}_{i=1}^{N}$ are learnable probabilities. The usage of a heuristic projection operator $\Pi_\mathcal{C}$ (see Appendix A for more details) as well as the KL divergence allows the theoretical convergence of categorical distribution RL under Cramér distance or Wasserstein distance (Rowland et al., 2018).

## 3 RISK-SENSITIVE REGULARIZATION IN DISTRIBUTIONAL RL

### 3.1 DISTRIBUTIONAL RL: NEURAL FZI

**Expectation-based RL: Neural Fitted Q-Iteration (Neural FQI).** Neural FQI (Fan et al., 2020; Riedmiller, 2005) offers a statistical explanation of DQN (Mnih et al., 2015), capturing its key features, including experience replay and the target network $Q_{\theta^*}$. In Neural FQI, we update a parameterized $Q_\theta$ in each iteration $k$ in a regression problem:

$$Q_\theta^{k+1} = \underset{Q_\theta}{\arg\min} \frac{1}{n} \sum_{i=1}^{n} [y_i - Q_\theta(s_i, a_i)]^2, \tag{1}$$

where the target $y_i = r(s_i, a_i) + \gamma \max_{a \in \mathcal{A}} Q_{\theta^*}^k(s_i', a)$ is fixed within every $T_{\text{target}}$ steps to update target network $Q_{\theta^*}$ by letting $Q_{\theta^*}^{k+1} = Q_\theta^{k+1}$. The experience buffer induces independent samples $\{(s_i, a_i, r_i, s_i')\}_{i \in [n]}$. If $\{Q_\theta : \theta \in \Theta\}$ is sufficiently large such that it contains $\mathcal{T}^{\text{opt}} Q_{\theta^*}^k$, Eq. 1 has solution $Q_\theta^{k+1} = \mathcal{T}^{\text{opt}} Q_{\theta^*}^k$, which is exactly the updating rule under Bellman optimality operator (Fan et al., 2020). In the viewpoint of statistics, the optimization problem in Eq. 1 in each iteration is a standard supervised and neural network parameterized regression regarding $Q_\theta$.

**Distributional RL: Neural Fitted Z-Iteration (Neural FZI).** We interpret distributional RL as Neural FZI as it is by far closest to the practical algorithms, although our analysis is not intended for involving properties of neural networks. Analogous to Neural FQI, we can simplify value-based distributional RL algorithms parameterized by $Z_\theta$ into Neural FZI as

$$Z_\theta^{k+1} = \underset{Z_\theta}{\arg\min} \frac{1}{n} \sum_{i=1}^n d_p(Y_i, Z_\theta(s_i, a_i)), \tag{2}$$

where the target $Y_i = R(s_i, a_i) + \gamma Z_{\theta^*}^k(s_i', \pi_Z(s_i'))$ with the policy $\pi_Z$ following the greedy rule $\pi_Z(s_i') = \arg\max_{a'} \mathbb{E}\left[Z_{\theta^*}^k(s_i', a')\right]$ is fixed within every $T_{\text{target}}$ steps to update target network $Z_{\theta^*}$. Here the lower cases random variables $s_i'$ and $\pi_Z(s_i')$ are given for convenience. $d_p$ is a divergence between two distributions.

## 3.2 Equivalent Form of Distributional RL: Entropy-regularized Neural FQI

**Return Density Function Decomposition.** To separate the impact of additional distribution information from the expectation of $Z^\pi$, we use a variant of *gross error model* from robust statistics (Huber, 2004), which was also similarly used to analyze Label Smoothing (Müller et al., 2019) and Knowledge Distillation (Hinton et al., 2015). Akin to the categorical representation in CDRL (Dabney et al., 2018b) we utilize a *histogram function estimator* $\widehat{p}^{s,a}(x)$ with $N$ bins to approximate an arbitrary continuous true action-value density $p^{s,a}(x)$ given a state $s$ and action $a$. We leverage the continuous histogram estimator rather than the discrete categorical parameterization to allow richer analysis. Given a fixed set of supports $l_0 \leq l_1 \leq ... \leq l_N$ with the equal bin size as $\Delta$, $\Delta_i = [l_{i-1}, l_i)$, $i = 1, ..., N-1$ with $\Delta_N = [l_{N-1}, l_N]$, the histogram density estimator is $\widehat{p}^{s,a}(x) = \sum_{i=1}^N p_i \mathbb{1}(x \in \Delta_i)/\Delta$ with the $i$-th bin height as $p_i/\Delta$. Denote $\Delta_E$ as the interval that $\mathbb{E}[Z^\pi(s,a)]$ falls into, i.e., $\mathbb{E}[Z^\pi(s,a)] \in \Delta_E$. Putting all together, we have an action-state return density function decomposition over the histogram density estimator $\widehat{p}^{s,a}(x)$:

$$\widehat{p}^{s,a}(x) = (1 - \epsilon)\mathbb{1}(x \in \Delta_E)/\Delta + \epsilon \widehat{\mu}^{s,a}(x) \tag{3}$$

where $\widehat{p}^{s,a}$ is decomposed into a single-bin histogram $\mathbb{1}(x \in \Delta_E)/\Delta$ and an *induced* histogram density function $\widehat{\mu}^{s,a}$ evaluated by $\widehat{\mu}^{s,a}(x) = \sum_{i=1}^N p_i^\mu \mathbb{1}(x \in \Delta_i)/\Delta$ with $p_i^\mu/\Delta$ as the $i$-th bin height. Optimizing the first term in Neural FZI is linked with Neural FQI for expectation-based RL, which we will show later. The induced histogram $\widehat{\mu}^{s,a}$ in the second term is to characterize the impact of action-state return distribution *despite its expectation* $\mathbb{E}[Z^\pi(s,a)]$ on the performance of distributional RL. $\epsilon$ is a pre-specified hyper-parameter before the decomposition, controlling the proportion between $\mathbb{1}(x \in \Delta_E)/\Delta$ and $\widehat{\mu}^{s,a}(x)$. Before establishing the equivalence between distributional RL and a specific entropy-regularized Neural FQI, we begin by showing that $\widehat{\mu}^{s,a}$ is a valid density function under certain $\epsilon$ in Proposition 1. The proof is provided in Appendix B.

**Proposition 1.** *(Decomposition Validity) Denote* $\widehat{p}^{s,a}(x \in \Delta_E) = p_E/\Delta$ *with* $p_E/\Delta$ *as the bin height.* $\widehat{\mu}^{s,a}(x) = \sum_{i=1}^N p_i^\mu \mathbb{1}(x \in \Delta_i)/\Delta$ *is a valid density function if and only if* $\epsilon \geq 1 - p_E$.

Proposition 1 indicates that the return density function decomposition in Eq. 3 is valid when the pre-specified hyper-parameter $\epsilon$ satisfies $\epsilon \geq 1 - p_E$, implying that $\epsilon \to 0$ is not attainable. Under this valid return density decomposition condition, this return density decomposition approach precisely maintains the standard categorical distribution framework in distributional RL.

**Histogram Function Parameterization Error: Uniform Convergence in Probability.** We further show that the histogram density estimator is equivalent to the categorical parameterization with the proof given in Appendix C, although the former is a continuous estimator in contrast to the discrete nature of the latter. However, the previous discrete categorical parameterization error bound in (Rowland et al., 2018) (Proposition 3) is derived between the true return distribution and the limiting return distribution denoted as $\eta_C$ iteratively updated via the Bellman operator $\Pi_C \mathfrak{T}^\pi$ *in expectation*, without considering an asymptotic analysis when the number of sampled $\{s_i, a_i\}_{i=1}^n$ pairs goes to infinity. As a complementary result, we provide a uniform convergence rate for the histogram density estimator in the context of distributional RL. In this particular analysis within this subsection, we denote $\widehat{p}_C^{s,a}$ as the density function estimator for the true limiting return distribution $\eta_C$ via $\Pi_C \mathfrak{T}^\pi$ with its true density $p_C^{s,a}$. In Theorem 1, we show that the sample-based histogram estimator $\widehat{p}_C^{s,a}$ can approximate any arbitrary continuous limiting density function $p_C^{s,a}$ under a mild condition. The proof is provided in Appendix D.

**Theorem 1.** *(Uniform Convergence Rate in Probability) Suppose $p_{\mathcal{C}}^{s,a}(x)$ is Lipschitz continuous and the support of a random variable is partitioned by $N$ bins with bin size $\Delta$. Then*

$$\sup_x |\widehat{p}_{\mathcal{C}}^{s,a}(x) - p_{\mathcal{C}}^{s,a}(x)| = O(\Delta) + O_P\left(\sqrt{\frac{\log N}{n\Delta^2}}\right). \tag{4}$$

**Distributional RL: Entropy-regularized Neural FQI.** We apply the decomposition on the target action-value histogram density function and choose KL divergence as $d_p$ in Neural FZI. Let $\mathcal{H}(U, V)$ be the cross-entropy between two probability measures $U$ and $V$, i.e., $\mathcal{H}(U, V) = -\int_{x \in \mathcal{X}} U(x) \log V(x)\, \mathrm{d}x$. The target histogram density function $\widehat{p}^{s,a}$ is decomposed as $\widehat{p}^{s,a}(x) = (1 - \epsilon)\mathbb{1}(x \in \Delta_E)/\Delta + \epsilon\widehat{\mu}^{s,a}(x)$. We can derive the following entropy-regularized form for distributional RL in Proposition 2. The proof is given in Appendix F.

**Proposition 2.** *(Decomposed Neural FZI) Denote $q_{\theta}^{s,a}(x)$ as the histogram estimator of $Z_{\theta}^k(s, a)$ in Neural FZI. Based on Eq. 3 and KL divergence as $d_p$, Neural FZI in Eq. 2 is simplified as*

$$Z_{\theta}^{k+1} = \underset{q_\theta}{\arg\min} \frac{1}{n} \sum_{i=1}^{n} [\underbrace{-\log q_\theta^{s_i,a_i}(\Delta_E^i)}_{(a)} + \alpha\mathcal{H}(\widehat{\mu}^{s_i',\pi_Z(s_i')}, q_\theta^{s_i,a_i})], \tag{5}$$

*where $\alpha = \varepsilon/(1 - \varepsilon) > 0$ and $\Delta_E^i$ represents the interval that the expectation of the target return distribution $R(s_i, a_i) + \gamma Z_{\theta^*}^k(s_i', \pi_Z(s_i'))$ falls into, i.e., $\mathbb{E}\left[R(s_i, a_i) + \gamma Z_{\theta^*}^k(s_i', \pi_Z(s_i'))\right] \in \Delta_E^i$. $\widehat{\mu}^{s_i',\pi_Z(s_i')}$ is the induced histogram density function by decomposing the histogram density estimator of $R(s_i, a_i) + \gamma Z_{\theta^*}^k(s_i', \pi_Z(s_i'))$ via Eq. 3. In Proposition 3, we further show that minimizing the term (a) in Eq. 5 is equivalent to minimizing Neural FQI, and therefore the regularization term $\alpha\mathcal{H}(\widehat{\mu}^{s_i',\pi_Z(s_i')}, q_\theta^{s_i,a_i})$ can be sufficiently used to interpret the benefits of CDRL over classical RL. For the uniformity of notation, we still use $s, a$ in the following analysis instead of $s_i, a_i$.*

**Proposition 3.** *(Equivalence between **the term (a)** in Decomposed Neural FZI and Neural FQI) In Eq. 5 of Neural FZI, if the function class $\{Z_\theta : \theta \in \Theta\}$ is sufficiently large such that it contains the target $\{Y_i\}_{i=1}^n$. As $\Delta \to 0$, for all $k$, minimizing **the term (a)** in Eq. 5 implies*

$$P(Z_\theta^{k+1}(s, a) = \mathcal{T}^{opt}Q_{\theta^*}^k(s, a)) = 1, \quad and \quad \int_{-\infty}^{+\infty} \left| F_{q_\theta}(x) - F_{\delta_{\mathcal{T}^{opt}Q_{\theta^*}^k(s,a)}}(x) \right| dx = o(\Delta), \tag{6}$$

*where $\delta_{\mathcal{T}^{opt}Q_{\theta^*}^k(s,a)}$ is the delta function defined on $\mathcal{T}^{opt}Q_{\theta^*}^k(s, a)$.*

The proof is given in Appendix G. Given the fact that $\{Z_\theta : \theta \in \Theta\}$ is sufficiently large such that it contains $\{Y_i\}_{i=1}^n$ in Neural FZI, we have $Z_\theta^{k+1} = \mathcal{T}^{opt}Q_{\theta^*}^k$ with probability one when $\Delta \to 0$. This result establishes a theoretical link between Neural FZI regarding the term (a) in Eq. 5 with Neural FQI, allowing us to leverage the regularization term $\alpha\mathcal{H}(\widehat{\mu}^{s_i',\pi_Z(s_i')}, q_\theta^{s_i,a_i})$ to explain the benefits of CDRL over classical RL. Next, we shift out attention to elaborating the impact of the regularization part in Eq. 5 for Neural FZI.

**Risk-Sensitive Entropy Regularization in Proposition 2.** Based on the equivalence between the term (a) of decomposed Neural FZI and FQI, we, therefore, interpret the form of distributional RL in Eq. 5 as *entropy-regularized Neural FQI*. As such, the behavior difference of distributional RL compared with expectation-based RL, especially the ability to significantly reduce intrinsic uncertainty of the environment (Mavrin et al., 2019), can be attributed to the second regularization term $\mathcal{H}(\widehat{\mu}^{s_i',\pi_Z(s_i')}, q_\theta^{s,a})$. It pushes $q_\theta^{s,a}$ for the current state-action pair to approximate $\widehat{\mu}^{s_i',\pi_Z(s_i')}$ for the target state-action pair, which additionally incorporates the return distribution information in the whole learning process instead of only encoding its expectation. According to the literature of risks in RL (Dabney et al., 2018a), where *"risk" refers to the uncertainty over possible outcomes and "risk-sensitive policies" are those which depend upon more than the mean of the outcomes*, we hereby call the novel cross-entropy regularization for the second term in Eq. 5 as *risk-sensitive entropy regularization*. This risk-sensitive entropy regularization derived within distributional RL expands the class of policies using the information provided by the distribution over returns (i.e. to the class of risk-sensitive policies).

**Remark on KL Divergence.** As stated in Section 2 of CDRL (Bellemare et al., 2017a), when the categorical parameterization is applied after the projection operator $\Pi_{\mathcal{C}}$, the distributional Bellman

operator $\mathfrak{T}^{\pi}$ has the contraction guarantee under Cramér distance (Rowland et al., 2018), albeit the use of a non-expansive KL divergence (Morimura et al., 2012). Similarly, our histogram density parameterization with the projection $\Pi_{\mathcal{C}}$ and KL divergence also enjoys a contraction property due to the equivalence between optimizing histogram function and categorical distribution analyzed in Appendix C. We summarize favorable properties of KL divergence in distributional RL in Appendix E.

**How to Obtain a Good Approximation of $\widehat{\mu}^{s',\pi_Z(s')}$?** As in practical distributional RL algorithms, we typically use the bootstrap, e.g., TD learning, to attain the target probability density estimate $\widehat{\mu}^{s',\pi_Z(s')}$ based on Eq. 3 as long as $\mathbb{E}\left[Z(s,a)\right]$ exists and $\epsilon \geq 1 - p_E$ in Proposition 1. The leverage of $\widehat{\mu}^{s',\pi_Z(s')}$ and the regularization effect revealed in Eq. 5 of distributional RL de facto establishes a bridge with MaxEnt RL (Williams & Peng, 1991) as analyzed in Section 4.

## 4 IMPLICIT REGULARIZATION IN THE ACTOR CRITIC FRAMEWORK

### 4.1 CONNECTION WITH MAXENT RL

**Explicit Vanilla Entropy Regularization in MaxEnt RL.** MaxEnt RL (Williams & Peng, 1991), including Soft Q-Learning (Haarnoja et al., 2017), *explicitly* encourages the exploration by optimizing for policies that aim to reach states where they will have high entropy in the future:

$$J(\pi) = \sum_{t=0}^{T} \mathbb{E}_{(s_t,a_t)\sim\rho_\pi}\left[r\left(s_t,a_t\right) + \beta\mathcal{H}(\pi(\cdot|s_t))\right], \tag{7}$$

where $\mathcal{H}\left(\pi_\theta\left(\cdot|s_t\right)\right) = -\sum_a \pi_\theta\left(a|s_t\right)\log\pi_\theta\left(a|s_t\right)$ and $\rho_\pi$ is the generated distribution following $\pi$. The temperature parameter $\beta$ determines the relative importance of the entropy term against the cumulative rewards and thus controls the action diversity of the optimal policy learned via Eq. 7. This maximum entropy regularization has various conceptual and practical advantages. Firstly, the learned policy is encouraged to visit states with high entropy in the future, thus promoting the exploration of diverse states (Han & Sung, 2021). It also considerably improves the learning speed (Mei et al., 2020) and therefore is widely used in state-of-the-art algorithms, e.g., Soft Actor-Critic (SAC) (Haarnoja et al., 2018). Similar empirical benefits of both distributional RL and MaxEnt RL motivate us to probe their underlying connection.

**Implicit Risk-Sensitive Entropy Regularization in Distributional RL.** To make a direct comparison with MaxEnt RL, we need to specifically analyze the impact of the regularization term in Eq. 5, and thus we incorporate the risk-sensitive entropy regularization of distributional RL into the Actor Critic (AC) framework akin to MaxEnt RL. We thus consider a new soft Q-value, i.e., the expectation of $Z^\pi(s,a)$. The new Q function can be computed iteratively by applying a modified Bellman operator $\mathcal{T}_d^\pi$ which we call *Distribution-Entropy-Regularized Bellman Operator* defined as

$$\mathcal{T}_d^\pi Q\left(s_t,a_t\right) \triangleq r\left(s_t,a_t\right) + \gamma\mathbb{E}_{s_{t+1}\sim P(\cdot|s_t,a_t)}\left[V\left(s_{t+1}|s_t,a_t\right)\right], \tag{8}$$

where a new soft value function $V\left(s_{t+1}|s_t,a_t\right)$ conditioned on $s_t, a_t$ is defined by

$$V\left(s_{t+1}|s_t,a_t\right) = \mathbb{E}_{a_{t+1}\sim\pi}\left[Q\left(s_{t+1},a_{t+1}\right)\right] + f(\mathcal{H}\left(\mu^{s_t,a_t},q_\theta^{s_t,a_t}\right)), \tag{9}$$

where $f$ is a continuous increasing function over the cross-entropy $\mathcal{H}$. Note that in this specific tabular setting regarding $s_t, a_t$, we particularly use $q_\theta^{s_t,a_t}(x)$ to approximate the true density function of $Z(s_t,a_t)$. We use $\mu^{s_t,a_t}$ to represent the induced true target return histogram function via the decomposition in Eq. 3 regardless of its expectation, which can typically be approximated via bootstrap estimate $\widehat{\mu}^{s_{t+1},\pi_Z(s_{t+1})}$ similar in Eq. 5. The $f$ transformation over the cross-entropy $\mathcal{H}$ between $\mu^{s_t,a_t}$ and $q_\theta^{s_t,a_t}(x)$ serves as the risk-sensitive entropy regularization that we implicitly derive from value-based distributional RL in Section 3.2. Here, we elaborate its impact on the optimization in actor-critic framework in contrast to MaxEnt RL.

**Implicit Reward Augmentation for a Different Exploration.** As opposed to the vanilla entropy regularization in MaxEnt RL that explicitly encourages the policy to explore, our risk-sensitive entropy regularization in distributional RL plays a role of the implicit **reward augmentation**. The augmented reward incorporates additional return distribution knowledge in the learning process compared with expectation-based RL. As suggested in Eq. 9, *the augmented reward encourages*

*policies to reach states $s_t$ with the following actions $a_t \sim \pi(\cdot|s_t)$, whose current action-state return distribution $q_\theta^{s_t,a_t}$ lag far behind the target ones, measured by the cross entropy.*

For a comprehensive analysis and a detailed comparison with MaxEnt RL, we now concentrate on the properties of our risk-sensitive entropy regularization in the framework of Actor Critic (AC). In Lemma 1, we first show that our Distribution-Entropy-Regularized Bellman operator $\mathcal{T}_d^\pi$ still inherits the convergence property in the policy evaluation phase with a cumulative augmented reward function as the new objective function.

**Lemma 1.** *(Distribution-Entropy-Regularized Policy Evaluation) Consider the distribution-entropy-regularized Bellman operator $\mathcal{T}_d^\pi$ in Eq. 8 and assume $\mathcal{H}(\mu^{s_t,a_t}, q_\theta^{s_t,a_t}) \leq M$ for all $(s_t, a_t) \in \mathcal{S} \times \mathcal{A}$, where $M$ is a constant. Define $Q^{k+1} = \mathcal{T}_d^\pi Q^k$, then $Q^{k+1}$ will converge to a corrected Q-value of $\pi$ as $k \to \infty$ with the new objective function $J'(\pi)$ defined as*

$$J'(\pi) = \sum_{t=0}^{T} \mathbb{E}_{(s_t,a_t) \sim \rho_\pi} \left[ r\left(s_t, a_t\right) + \gamma f(\mathcal{H}\left(\mu^{s_t,a_t}, q_\theta^{s_t,a_t}\right)) \right]. \tag{10}$$

In the policy improvement for distributional RL, we keep the vanilla updating rules according to $\pi_{\text{new}} = \arg\max_{\pi' \in \Pi} \mathbb{E}_{a_t \sim \pi'} \left[ Q^{\pi_{\text{old}}}(s_t, a_t) \right]$. Next, we can immediately derive a new policy iteration algorithm, called *Distribution-Entropy-Regularized Policy Iteration (DERPI)* that alternates between the policy evaluation in Eq. 8 and the policy improvement. It will provably converge to the policy with the optimal risk-sensitive entropy among all policies in $\Pi$ as shown in Theorem 2.

**Theorem 2.** *(Distribution-Entropy-Regularized Policy Iteration) Assume $\mathcal{H}(\mu^{s_t,a_t}, q_\theta^{s_t,a_t}) \leq M$ for all $(s_t, a_t) \in \mathcal{S} \times \mathcal{A}$, where $M$ is a constant. Repeatedly applying distribution-entropy-regularized policy evaluation in Eq. 8 and the policy improvement, the policy converges to an optimal policy $\pi^*$ such that $Q^{\pi^*}(s_t, a_t) \geq Q^\pi(s_t, a_t)$ for all $\pi \in \Pi$.*

Please refer to Appendix H for the proof of Lemma 1 and Theorem 2. Theorem 2 indicates that if we incorporate the risk-sensitive entropy regularization into the policy gradient framework in Eq. 10, we can design a variant of "soft policy iteration" (Haarnoja et al., 2018) that can guarantee the convergence to an optimal policy, where the optimal policy is defined based on the optimal Q function. Based on the analysis above, we next provide a comprehensive comparison between the explicit vanilla entropy in MaxEnt RL and the implicit risk-sensitive entropy in distributional RL.

**Explicit vs Implicit Policy Optimization and Exploration.** By comparing $J(\pi)$ in Eq. 7 and $J'(\pi)$ in Eq. 10, the state-wise entropy $\mathcal{H}(\pi(\cdot|s_t))$ is maximized explicitly *w.r.t.* $\pi$ in MaxEnt RL for policies with a higher entropy in terms of diverse actions. In contrast, distributional RL implicitly maximizes the risk-sensitive entropy regularization *w.r.t.* $\pi$ via $a_t \sim \pi(\cdot|s_t)$, leading to different impact of exploration. Concretely, the learned policy is encouraged to *visit state $s_t$ along with the policy-determined action pairs via $a_t \sim \pi(\cdot|s_t)$ in the future whose current action-state return distributions $q_\theta^{s_t,a_t}$ "lag far behind" compared with the target return distributions, measure by the cross entropy.* In expectation-based RL, the learned $q_\theta^{s_t,a_t}$ is more likely to concentrate on the expectation of target return distribution, without the

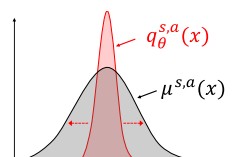

Figure 1: Intrinsic uncertainty reduction via risk-sensitive exploration. $q_\theta^{s,a}$ is encouraged to disperse under the risk-sensitive entropy regularization.

leverage of the full return distribution information. Thus, optimizing the implicit regularization in distributional RL pushes $q_\theta^{s_t,a_t}$ to approach the target return distribution $\mu^{s_t,a_t}$ that tends to have a higher degree of dispersion, e.g., variance. As such, the implicit risk-sensitive entropy potentially promotes the **risk-sensitive exploration** to reduce the intrinsic uncertainty of the environment, as illustrated in Figure 1 as an example. It is still possible that $q_\theta^{s_t,a_t}$ has already a higher variance than $\mu^{s_t,a_t}$ in the learning process and thus it is to be optimized to reduce the dispersion. We argue that it highly depends on the environment and learning phases to determine which scenario happens.

## 4.2 DERAC Algorithm: Interpolating AC and Distributional AC

For a practical algorithm, we extend DERPI to the function approximation setting by parameterizing the return distribution $q_\theta(s_t, a_t)$ and the policy $\pi_\phi(a_t|s_t)$, yielding the Distribution-Entropy-Regularized Actor-Critic (DERAC) that interpolates expectation-based AC and distributional AC.

**Optimize the parameterized return distribution** $q_\theta$**.** The new value function $\hat{J}_q(\theta)$ is originally trained to minimize the squared residual error of Eq. 8. We show that $\hat{J}_q(\theta)$ can be simplified as:

$$\hat{J}_q(\theta) \propto (1-\lambda)\mathbb{E}_{s,a}\left[(\mathcal{T}^\pi \mathbb{E}\left[q_{\theta^*}(s,a)\right] - \mathbb{E}\left[q_\theta(s,a)\right])^2\right] + \lambda\mathbb{E}_{s,a}\left[\mathcal{H}(\mu^{s,a}, q_\theta^{s,a})\right], \tag{11}$$

where we use a particular increasing function $f(\mathcal{H}) = (\tau\mathcal{H})^{\frac{1}{2}}/\gamma$ and $\lambda = \frac{\tau}{1+\tau} \in [0,1], \tau \geq 0$ is the hyperparameter that controls the risk-sensitive regularization effect. The proof is given in Appendix I. Interestingly, when we leverage the whole target density function $\hat{p}^{s,a}$ to approximate the true $\mu^{s,a}$, the objective function in Eq. 11 can be viewed as an exact interpolation of loss functions between expectation-based AC (the first term) and categorical distributional AC loss (the second term)(Ma et al., 2020). Note that for the target $\mathcal{T}^\pi \mathbb{E}\left[q_{\theta^*}(s,a)\right]$, we use the target return distribution neural network $q_{\theta^*}$ to stabilize the training, which is consistent with the Neural FZI framework analyzed in Section 3.1.

**Optimize the policy** $\pi_\phi$**.** We optimize $\pi_\phi$ in the policy optimization based on the Q-function and therefore the new objective function $\hat{J}_\pi(\phi)$ can be expressed as $\hat{J}_\pi(\phi) = \mathbb{E}_{s,a\sim\pi_\phi}\left[\mathbb{E}\left[q_\theta(s,a)\right]\right]$. The complete DERAC algorithm is described in Algorithm 1 of Appendix K.

## 5 EXPERIMENTS

In our experiments, we first verify the risk-sensitive entropy regularization effect in value-based CDRL analyzed in Section 3 on eight typical Atari games. For the actor-critic framework analyzed in Section 4, we demonstrate the implicit regularization in Distributional SAC (DSAC) (Ma et al., 2020) with C51 as the critic loss, as well as the interpolation behavior of DERAC algorithm in continuous control environments. Finally, an empirical extension to Implicit Quantile Networks (IQN) (Dabney et al., 2018a) is provided on eight MuJoCo environments to reveal the mutual impacts of explicit and implicit entropy regularization. The implementation of the DERAC algorithm is based on DSAC (Ma et al., 2020), which also serves as a baseline. More implementation details are provided in Appendix J.

### 5.1 RISK-SENSITIVE ENTROPY REGULARIZATION IN VALUE-BASED DISTRIBUTIONAL RL

We demonstrate the rationale of action-state return density function decomposition in Eq. 3 and the risk-sensitive entropy regularization effect analyzed in Eq. 5 based on the C51 algorithm. Firstly, it is a fact that the return distribution decomposition is based on the equivalence between KL divergence and cross-entropy owing to the usage of target networks. Hence, we demonstrate that the C51 algorithm can still achieve similar results under the cross-entropy loss across both Atari games and MuJoCo environments in Figure 5 of Appendix L. In the value-based C51 loss, we replace the whole target categorical distribution $\hat{p}^{s,a}(x)$ in C51 with the derived $\hat{\mu}^{s,a}(x)$ under different $\varepsilon$ based on Eq. 3 in the cross-entropy loss, allowing to investigate the risk-sensitive regularization effect of

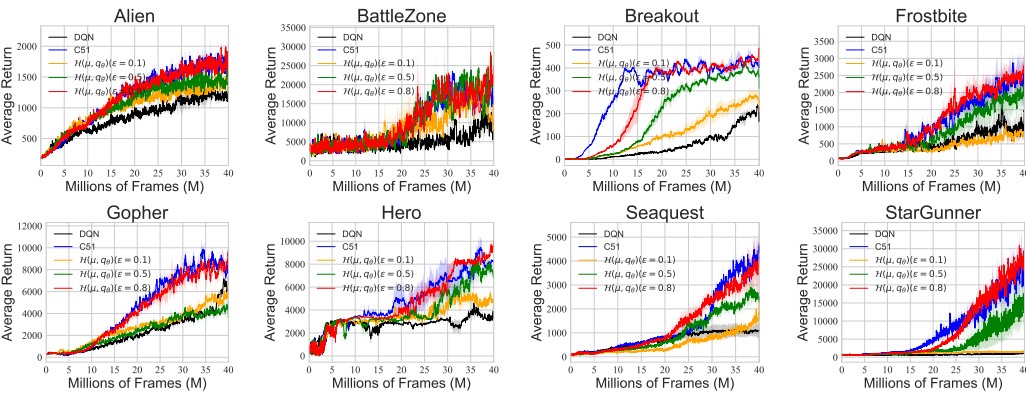

Figure 2: Learning curves of value-based CDRL, i.e., C51 algorithm, with return distribution decomposition $\mathcal{H}(\mu, q_\theta)$ under different $\varepsilon$ on eight typical Atari games averaged over 3 seeds.

distributional RL. Concretely, we define $\varepsilon$ as the proportion of probability of the bin that contains the expectation *with mass to be transported to other bins*. We use $\varepsilon$ to replace $\epsilon$ for convenience as the leverage of $\varepsilon$ can always guarantee the valid density function $\widehat{\mu}$ analyzed in Proposition 1. A large proportion probability $\varepsilon$ that transports less mass to other bins corresponds to a large $\epsilon$ in Eq. 3, which would be closer to a distributional RL algorithm, i.e., C51.

As shown in Figure 2, when $\varepsilon$ gradually decreases from 0.8 to 0.1, learning curves of decomposed C51 denoted as $\mathcal{H}(\mu, q_\theta)(\varepsilon = 0.8/0.5/0.1)$ tend to degrade from vanilla C51 to DQN across most eight Atari games, although their sensitivity in terms of $\varepsilon$ may depend on the environment. This empirical observation corroborates the role of risk-sensitive entropy regularization we derive in Section 3.2, suggesting that the risk-sensitive entropy regularization is pivotal to the success of CDRL.

## 5.2 RISK-SENSITIVE ENTROPY REGULARIZATION IN CONTINUOUS CONTROL

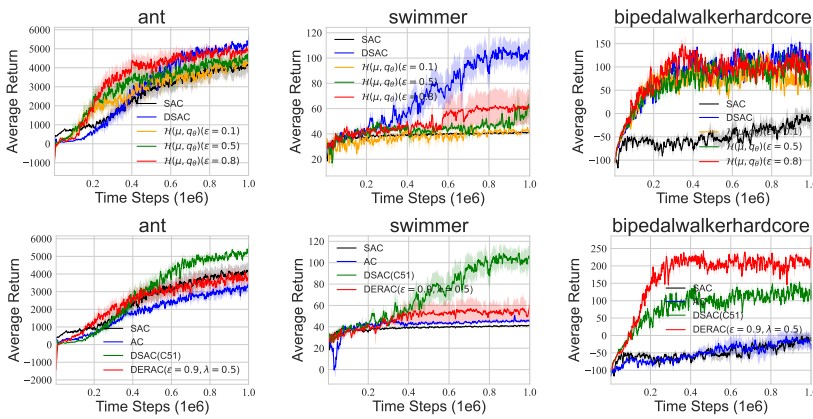

Figure 3: (**First Row**) Learning curves of Distributional AC (C51) with the return distribution decomposition $\mathcal{H}(\mu, q_\theta)$ under different $\varepsilon$. (**Second Row**) Learning curves of DERAC algorithm. All results are averaged over 5 seeds. We denote AC as SAC without the leverage of entropy.

As suggested in the first row of Figure 3, the performance of the decomposed DSAC (C51), denoted as $\mathcal{H}(\mu, q_\theta)(\varepsilon = 0.8/0.5/0.1)$, also tends to vary from the vanilla DSAC (C51) to SAC with the decreasing of $\varepsilon$ on three MuJoCo environments, except bipedalwalkerhardcore. This is because our return density decomposition is valid only when $\epsilon \geq 1 - p_E$ as shown in Proposition 1, and therefore $\epsilon$ can not strictly go to 0, where $\mathcal{H}(\mu, q_\theta)$ would degenerate to SAC ideally.

The second row of Figure 3 showcases that DERAC (the red line) tends to "interpolate" between the expectation-based AC (without vanilla entropy) / SAC and DSAC (C51) across three MuJoCo environments, except bipedalwalkerhardcore (hard for exploration), where the interpolation has extra advantages. We hypothesize that this risk-sensitive regularization is more likely to improve the performance on complicated environments, e.g., bipedalwalkerhardcore, for which we provide more results and discussions in Appendix M.

*We emphasize that introducing the DERAC algorithm is not to pursue the empirical outperformance over DSAC but to corroborate the rationale of incorporating risk-sensitive entropy regularization in actor-critic framework*, including the theoretical convergence of the tabular DERPI algorithm in Theorem 2, by observing the interpolating behavior of DERAC between SAC and DSAC. Specifically, as we choose $\varepsilon = 0.9$ in DERAC, there exists a distribution information loss, resulting in the learning performance degradation, e.g., on Swimmer. To pursue the performance in practice, our suggestion is to directly deploy DSAC that takes advantage of the full return distribution information. We also provide a sensitivity analysis of DERAC regarding $\lambda$ in Figure 6 of Appendix L.

## 5.3 MUTUAL IMPACTS OF EXPLICIT AND IMPLICIT REGULARIZATION AND BEYOND

Since the implicit regularization we reveal is highly linked to CDRL, we study the mutual impacts of explicit regularization in SAC and implicit regularization in DSAC in quantile-based distributional RL, e.g., QR-DQN (Dabney et al., 2018b) to reveal that the impact regularization in CDRL

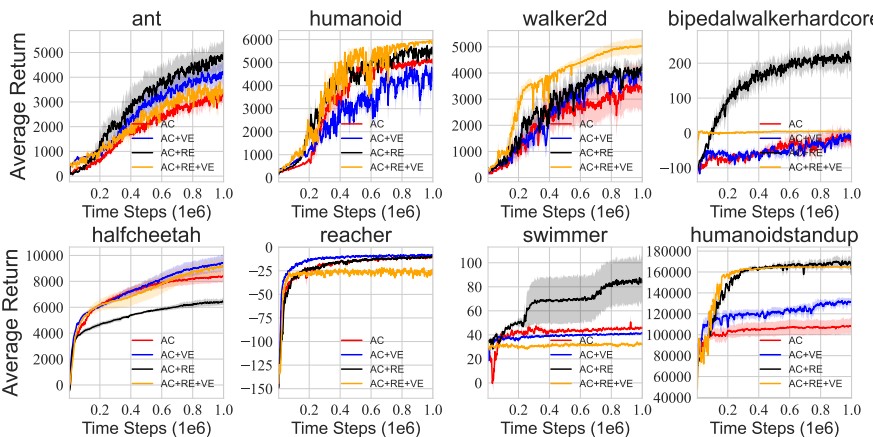

Figure 4: Learning curves of AC, AC+VE (SAC), AC+RE and AC+RE+VE (DSAC) over 5 seeds with smooth size 5 across eight MuJoCo environments where distributional RL is based on IQN.

potentially exists in the general distributional RL. Similar results conducted on DSAC (C51) are also given in Appendix N. Specifically, we conduct a careful ablation study to control the effects of vanilla entropy (VE), risk-sensitive entropy (RE), and their mutual impacts. We denote SAC with and without vanilla entropy as *AC* and *AC+VE*, and distributional SAC with and without vanilla entropy as *AC+RE+VE* and *AC+RE*, where VE and RE are short for *Vanilla Entropy* and *Risk-sensitive Entropy*. For the implementation, we leverage the quantiles generation strategy in IQN (Dabney et al., 2018a) in distributional SAC (Ma et al., 2020). Hyper-parameters are listed in Appendix J. As suggested in Figure 4, we make the following conclusions:

**(1)** With no surprise, the explicit vanilla entropy is useful as AC+VE (blue line) outperforms AC (red lines) across most environments except on the humanoid and swimmer. By contrast, The implicit risk-sensitive entropy effect (RE) from distributional RL also benefits the learning as AC+RE (black lines) tends to bypass AC (red lines), especially on the complex BipealWalkerHardcore (hard for exploration) and Humanoidstandup (with the specific objective as opposed to Humanoid).

**(2)** The leverage of both risk-sensitive entropy and vanilla entropy may interfere with each other, e.g., on BipealWalkerHardcore and Swimmer games, where *AC+RE+VE* (orange lines) is significantly inferior to *AC+RE* (black lines). This may result from the different policy optimization/exploration preferences of two regularization effects. SAC explicitly optimizes the policy to visit states with high entropy, while distributional RL implicitly optimizes the policy to visit states and the following actions whose return distribution has a higher cross-entropy for the current return distribution estimate, thus potentially promoting the risk-sensitive exploration. We hypothesize that mixing two different policy optimization/exploration directions may lead to sub-optimal solutions in certain environments, thus interfering with each other eventually.

## 6    DISCUSSIONS AND CONCLUSION

The implicit regularization effect we reveal is mainly based on CDRL. Although CDRL is viewed as the first successfully distributional RL family, the theoretical techniques, including the contraction analysis, in other distributional RL families, e.g., QR-DQN, are highly different from CDRL (Rowland et al., 2023). Hence, there remain some theoretical gaps to extend the implicit regularization conclusions in CDRL to general distributional RL algorithms, which we leave as future work.

In this paper, we interpret the potential superiority of CDRL over expectation-based RL as the implicit regularization derived through the return density decomposition. In contrast to the explicit policy optimization in MaxEnt RL, the risk-sensitive regularization in CDRL serves as an augmented reward, which implicitly optimizes the policy. Starting from CDRL, our research contributes to a deeper understanding of the potential superiority of distributional RL algorithms.

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

## A    Convergence Guarantee of Categorical Distributional RL

Categorical Distributional RL (Bellemare et al., 2017a) uses the heuristic projection operator $\Pi_{\mathcal{C}}$ that was defined as

$$\Pi_{\mathcal{C}}\left(\delta_y\right) = \begin{cases} \delta_{l_1} & y \le l_1 \\ \frac{l_{i+1}-y}{l_{i+1}-z_i}\delta_{l_i} + \frac{y-l_i}{l_{i+1}-z_i}\delta_{l_{i+1}} & l_i < y \le l_{i+1} \\ \delta_{l_K} & y > l_K \end{cases}, \tag{12}$$

and extended affinely to finite mixtures of Dirac measures, so that for a mixture of Diracs $\sum_{i=1}^{N} p_i \delta_{y_i}$, we have $\Pi_{\mathcal{C}}\left(\sum_{i=1}^{N} p_i \delta_{y_i}\right) = \sum_{i=1}^{N} p_i \Pi_{\mathcal{C}}\left(\delta_{y_i}\right)$. The Cramér distance was recently studied as an alternative to the Wasserstein distances in the context of generative models (Bellemare et al., 2017b). Recall the definition of Cramér distance.

**Definition 1.** *(Definition 3 (Rowland et al., 2018)) The Cramér distance $\ell_2$ between two distributions $\nu_1, \nu_2 \in \mathscr{P}(\mathbb{R})$, with cumulative distribution functions $F_{\nu_1}, F_{\nu_2}$ respectively, is defined by:*

$$\ell_2\left(\nu_1, \nu_2\right) = \left(\int_{\mathbb{R}} \left(F_{\nu_1}(x) - F_{\nu_2}(x)\right)^2 \, \mathrm{d}x\right)^{1/2}.$$

*Further, the supremum-Cramér metric $\bar{\ell}_2$ is defined between two distribution functions $\eta, \mu \in \mathscr{P}(\mathbb{R})^{\mathcal{X} \times \mathcal{A}}$ by*

$$\bar{\ell}_2(\eta, \mu) = \sup_{(x,a) \in \mathcal{X} \times \mathcal{A}} \ell_2\left(\eta^{(x,a)}, \mu^{(x,a)}\right).$$

Thus, the contraction of categorical distributional RL can be guaranteed under Cramér distance:

**Proposition 4.** *(Proposition 2 (Rowland et al., 2018)) The operator $\Pi_{\mathcal{C}}\mathcal{T}^{\pi}$ is a $\sqrt{\gamma}$-contraction in $\bar{\ell}_2$.*

An insight behind this conclusion is that Cramér distance endows a particular subset with a notion of orthogonal projection, and the orthogonal projection onto the subset is exactly the heuristic projection $\Pi_{\mathcal{C}}$ (Proposition 1 in (Rowland et al., 2018)).

## B    Proof of Proposition 1

**Proposition 1.** Denote $\widehat{p}^{s,a}(x \in \Delta_E) = p_E/\Delta$. Following the density function decomposition in Eq. 3, $\widehat{\mu}(x) = \sum_{i=1}^{N} p_i^{\mu} \mathbb{1}(x \in \Delta_i)/\Delta$ is a valid probability density function if and only if $\epsilon \ge 1 - p_E$.

*Proof.* Recap a valid probability density function requires non-negative and one-bounded probability in each bin and all probabilities should sum to 1.

**Necessity.** (1) When $x \in \Delta_E$, Eq. 3 can simplified as $p_E/\Delta = (1-\epsilon)/\Delta + \epsilon p_E^{\mu}/\Delta$, where $p_E^{\mu} = \widehat{\mu}(x \in \Delta_E)$. Thus, $p_E^{\mu} = \frac{p_E}{\epsilon} - \frac{1-\epsilon}{\epsilon} \ge 0$ if $\epsilon \ge 1 - p_E$. Obviously, $p_E^{\mu} = \frac{p_E}{\epsilon} - \frac{1-\epsilon}{\epsilon} \le \frac{1}{\epsilon} - \frac{1-\epsilon}{\epsilon} = 1$ guaranteed by the validity of $\widehat{p}^{s,a}$. (2) When $x \notin \Delta_E$, we have $p_i/\Delta = \epsilon p_i^{\mu}/\Delta$, i.e., When $x \notin \Delta_E$, We immediately have $p_i^{\mu} = \frac{p_i}{\epsilon} \le \frac{1-p_E}{\epsilon} \le 1$ when $\epsilon \ge 1 - p_E$. Also, $p_i^{\mu} = \frac{p_i}{\epsilon} \ge 0$.

**Sufficiency.** (1) When $x \in \Delta_E$, let $p_E^{\mu} = \frac{p_E}{\epsilon} - \frac{1-\epsilon}{\epsilon} \ge 0$, we have $\epsilon \ge 1 - p_E$. $p_E^{\mu} = \frac{p_E}{\epsilon} - \frac{1-\epsilon}{\epsilon} \le 1$ in nature. (2) When $x \notin \Delta_E$, $p_i^{\mu} = \frac{p_i}{\epsilon} \ge 0$ in nature. Let $p_i^{\mu} = \frac{p_i}{\epsilon} \le 1$, we have $p_i \le \epsilon$. We need to take the intersection set of (1) and (2), and we find that $\epsilon \ge 1 - p_E \Rightarrow \epsilon \ge 1 - p_E \ge p_i$ that satisfies the condition in (2). Thus, the intersection set of (1) and (2) would be $\epsilon \ge 1 - p_E$.

In summary, as $\epsilon \ge 1 - p_E$ is both the necessary and sufficient condition, we have the conclusion that $\widehat{\mu}(x)$ is a valid probability density function $\iff \epsilon \ge 1 - p_E$.

$\square$

## C  EQUIVALENCE BETWEEN CATEGORICAL AND HISTOGRAM PARAMETERIZATION

**Proposition 5.** *Suppose the target categorical distribution $c = \sum_{i=1}^{N} p_i \delta_{z_i}$ and the target histogram function $h(x) = \sum_{i=1}^{N} p_i \mathbb{1}(x \in \Delta_i)/\Delta$, updating the parameterized categorical distribution $c_\theta$ under KL divergence is equivalent to updating the parameterized histogram function $h_\theta$.*

*Proof.* For the histogram density estimator $h_\theta$ and the true target density function $p(x)$, we can simplify the KL divergence as follows.

$$
\begin{aligned}
D_{\mathrm{KL}}(h, h_\theta) &= \sum_{i=1}^{N} \int_{l_{i-1}}^{l_i} \frac{p_i(x)}{\Delta} \log \frac{\frac{p_i(x)}{\Delta}}{\frac{h_\theta^i}{\Delta}} dx \\
&= \sum_{i=1}^{N} \int_{l_{i-1}}^{l_i} \frac{p_i(x)}{\Delta} \log \frac{p_i(x)}{\Delta} dx - \sum_{i=1}^{N} \int_{l_{i-1}}^{l_i} \frac{p_i(x)}{\Delta} \log \frac{h_\theta^i}{\Delta} dx \\
&\propto - \sum_{i=1}^{N} \int_{l_{i-1}}^{l_i} \frac{p_i(x)}{\Delta} \log \frac{h_\theta^i}{\Delta} dx \\
&= - \sum_{i=1}^{N} p_i(x) \log \frac{h_\theta^i}{\Delta} \propto - \sum_{i=1}^{N} p_i(x) \log h_\theta^i
\end{aligned}
\tag{13}
$$

where $h_\theta^i$ is determined by $i$ and $\theta$ and is independent of $x$. For categorical distribution estimator $c_\theta$ with the probability $p_i$ in for each atom $z_i$, we also have its target categorical distribution $p(x)$ with each probability $p_i$, we have:

$$
\begin{aligned}
D_{\mathrm{KL}}(c, c_\theta) &= \sum_{i=1}^{N} p_i \log \frac{p_i}{c_\theta^i} \\
&= \sum_{i=1}^{N} p_i \log p_i - \sum_{i=1}^{N} p_i \log c_\theta^i \\
&\propto - \sum_{i=1}^{N} p_i \log c_\theta^i
\end{aligned}
\tag{14}
$$

$\square$

In CDRL, we only use a discrete categorical distribution with probabilities centered on the fixed atoms $\{z_i\}_{i=1}^{N}$, while the histogram density estimator in our analysis is a continuous function defined on $[z_0, z_N]$ to allow richer analysis. We reveal that minimizing the KL divergence regarding the parameterized categorical distribution in Eq. 14 is equivalent to minimizing the cross-entropy loss regarding the parameterized histogram function in Eq. 13.

## D  PROOF OF THEOREM 1

**Theorem 1**. Suppose $p_{\mathcal{C}}^{s,a}(x)$ is Lipschitz continuous and the support of $X$ is partitioned by N bins with bin size $\Delta$. Then

$$
\sup_x |\widehat{p}_{\mathcal{C}}^{s,a}(x) - p_{\mathcal{C}}^{s,a}(x)| = O(\Delta) + O_P\left( \sqrt{\frac{\log N}{n\Delta^2}} \right).
\tag{15}
$$

*Proof.* Our proof is mainly based on the non-parametric statistics analysis (Wasserman, 2006). In particular, the difference of $\widehat{p}_{\mathcal{C}}^{s,a}(x) - p_{\mathcal{C}}^{s,a}(x)$ can be written as

$$
\widehat{p}_{\mathcal{C}}^{s,a}(x) - p_{\mathcal{C}}^{s,a}(x) = \underbrace{\mathbb{E}\left(\widehat{p}_{\mathcal{C}}^{s,a}(x)\right) - p_{\mathcal{C}}^{s,a}(x)}_{\text{bias}} + \underbrace{\widehat{p}_{\mathcal{C}}^{s,a}(x) - \mathbb{E}\left(\widehat{p}_{\mathcal{C}}^{s,a}(x)\right)}_{\text{stochastic variation}}.
\tag{16}
$$

**(1) The first bias term.** Without loss of generality, we consider $x \in \Delta_k$, we have

$$
\begin{aligned}
\mathbb{E}\left(\widehat{p}_\mathcal{C}^{s,a}(x)\right) &= \frac{P(X \in \Delta_k)}{\Delta} \\
&= \frac{\int_{l_0+(k-1)\Delta}^{l_0+k\Delta} p(y)dy}{\Delta} \\
&= \frac{F(l_0 + (k-1)\Delta) - F(l_0 + (k-1)\Delta)}{l_0 + k\Delta - (l_0 + (k-1)\Delta)} \\
&= p_\mathcal{C}^{s,a}(x'),
\end{aligned}
\tag{17}
$$

where the last equality is based on the mean value theorem. According to the L-Lipschitz continuity property, we have

$$
\begin{aligned}
|\mathbb{E}\left(\widehat{p}_\mathcal{C}^{s,a}(x)\right) - p_\mathcal{C}^{s,a}(x)| &= |p_\mathcal{C}^{s,a}(x') - p_\mathcal{C}^{s,a}(x)| \\
&\leq L|x' - x| \\
&\leq L\Delta
\end{aligned}
\tag{18}
$$

**(2) The second stochastic variation term.** If we let $x \in \Delta_k$, then $\widehat{p}_\mathcal{C}^{s,a} = p_k = \frac{1}{n}\sum_{i=1}^n \mathbb{1}(X_i \in \Delta_k)$, we thus have

$$
\begin{aligned}
&P\left(\sup_x |\widehat{p}_\mathcal{C}^{s,a}(x) - \mathbb{E}\left(\widehat{p}_\mathcal{C}^{s,a}(x)\right)| > \epsilon\right) \\
&= P\left(\max_{j=1,\cdots,N} \left|\frac{1}{n}\sum_{i=1}^n \mathbb{1}\left(X_i \in \Delta_j\right)/\Delta - P\left(X_i \in \Delta_j\right)/\Delta\right| > \epsilon\right) \\
&= P\left(\max_{j=1,\cdots,N} \left|\frac{1}{n}\sum_{i=1}^n \mathbb{1}\left(X_i \in \Delta_j\right) - P\left(X_i \in \Delta_j\right)\right| > \Delta\epsilon\right) \\
&\leq \sum_{j=1}^N P\left(\left|\frac{1}{n}\sum_{i=1}^n \mathbb{1}\left(X_i \in \Delta_j\right) - P\left(X_i \in \Delta_j\right)\right| > \Delta\epsilon\right) \\
&\leq N \cdot \exp\left(-2n\Delta^2\epsilon^2\right) \quad \text{(by Hoeffding's inequality)},
\end{aligned}
\tag{19}
$$

where in the last inequality we know that the indicator function is bounded in [0, 1]. We then let the last term be a constant independent of $N, n, \Delta$ and simplify the order of $\epsilon$. Then, we have:

$$
\sup_x |\widehat{p}_\mathcal{C}^{s,a}(x) - \mathbb{E}\left(\widehat{p}_\mathcal{C}^{s,a}(x)\right)| = O_P\left(\sqrt{\frac{\log N}{n\Delta^2}}\right)
\tag{20}
$$

In summary, as the above inequality holds for each $x$, we thus have the uniform convergence rate of a histogram density estimator

$$
\begin{aligned}
\sup_x |\widehat{p}_\mathcal{C}^{s,a}(x) - p_\mathcal{C}^{s,a}(x)| &\leq \sup_x |\mathbb{E}\left(\widehat{p}_\mathcal{C}^{s,a}(x)\right) - p_\mathcal{C}^{s,a}(x)| + \sup_x |\widehat{p}_\mathcal{C}^{s,a}(x) - \mathbb{E}\left(\widehat{p}_\mathcal{C}^{s,a}(x)\right)| \\
&= O\left(\Delta\right) + O_P\left(\sqrt{\frac{\log N}{n\Delta^2}}\right).
\end{aligned}
\tag{21}
$$

$\square$

# E   PROPERTIES OF KL DIVERGENCE IN DISTRIBUTIONAL RL

**Proposition 6.** *Given two probability measures $\mu$ and $\nu$, we define the supreme $D_{KL}$ as a functional $\mathcal{P}(\mathcal{X})^{\mathcal{S}\times\mathcal{A}} \times \mathcal{P}(\mathcal{X})^{\mathcal{S}\times\mathcal{A}} \to \mathbb{R}$, i.e., $D_{KL}^\infty(\mu,\nu) = \sup_{(x,a)\in\mathcal{S}\times\mathcal{A}} D_{KL}(\mu(x,a),\nu(x,a))$. we have: (1) $\mathfrak{T}^\pi$ is a non-expansive distributional Bellman operator under $D_{KL}^\infty$, i.e., $D_{KL}^\infty(\mathfrak{T}^\pi Z_1, \mathfrak{T}^\pi Z_2) \leq D_{KL}^\infty(Z_1, Z_2)$, (2) $D_{KL}^\infty(Z_n, Z) \to 0$ implies the Wasserstein distance $W_p(Z_n, Z) \to 0$, (3) the expectation of $Z^\pi$ is still $\gamma$-contractive under $D_{KL}^\infty$, i.e., $\|\mathbb{E}\mathfrak{T}^\pi Z_1 - \mathbb{E}\mathfrak{T}^\pi Z_2\|_\infty \leq \gamma \|\mathbb{E}Z_1 - \mathbb{E}Z_2\|_\infty$.*

*Proof.* We firstly assume $Z_\theta$ is absolutely continuous and the supports of two distributions in KL divergence have a negligible intersection (Arjovsky & Bottou, 2017), under which the KL divergence is well-defined.

(1) Please refer to (Morimura et al., 2012) for the proof. Therefore, we have $D_{\mathrm{KL}}^\infty(\mathfrak{T}^\pi Z_1, \mathfrak{T}^\pi Z_2) \le D_{\mathrm{KL}}^\infty(Z_1, Z_2)$, implying that $\mathfrak{T}^\pi$ is a non-expansive operator under $D_{\mathrm{KL}}^\infty$.

(2) By the definition of $D_{\mathrm{KL}}^\infty$, we have $\sup_{s,a} D_{\mathrm{KL}}(Z_n(s,a), Z(s,a)) \to 0$ implies $D_{\mathrm{KL}}(Z_n, Z) \to 0$. $D_{\mathrm{KL}}(Z_n, Z) \to 0$ implies the total variation distance $\delta(Z_n, Z) \to 0$ according to a straightforward application of Pinsker's inequality

$$
\begin{aligned}
\delta(Z_n, Z) &\le \sqrt{\frac{1}{2} D_{\mathrm{KL}}(Z_n, Z)} \to 0 \\
\delta(Z, Z_n) &\le \sqrt{\frac{1}{2} D_{\mathrm{KL}}(Z, Z_n)} \to 0
\end{aligned}
\tag{22}
$$

Based on Theorem 2 in WGAN (Arjovsky et al., 2017), $\delta(Z_n, Z) \to 0$ implies $W_p(Z_n, Z) \to 0$. This is trivial by recalling the fact that $\delta$ and $W$ give the strong and weak topologies on the dual of $(C(\mathcal{X}), \|\cdot\|_\infty)$ when restricted to $\mathrm{Prob}(\mathcal{X})$.

(3) The conclusion holds because the $\mathfrak{T}^\pi$ degenerates to $\mathcal{T}^\pi$ regardless of the metric $d_p$ (Bellemare et al., 2017a). Specifically, due to the linearity of expectation, we obtain that

$$
\|\mathbb{E}\mathfrak{T}^\pi Z_1 - \mathbb{E}\mathfrak{T}^\pi Z_2\|_\infty = \|\mathcal{T}^\pi \mathbb{E} Z_1 - \mathcal{T}^\pi \mathbb{E} Z_2\|_\infty \le \gamma \|\mathbb{E} Z_1 - \mathbb{E} Z_2\|_\infty.
\tag{23}
$$

This implies that the expectation of $Z$ under $D_{\mathrm{KL}}$ exponentially converges to the expectation of $Z^*$, i.e., $\gamma$-contraction. $\square$

## F  PROOF OF PROPOSITION 2

**Proposition 2** (Decomposed Neural FZI) Denote $q_\theta^{s,a}(x)$ as the histogram density function of $Z_\theta^k(s,a)$ in Neural FZI. Based on Eq. 3 and KL divergence as $d_p$, Neural FZI in Eq. 2 is simplified as

$$
Z_\theta^{k+1} = \underset{q_\theta}{\arg\min} \frac{1}{n} \sum_{i=1}^n [\underbrace{-\log q_\theta^{s_i, a_i}(\Delta_E^i)}_{(a)} + \alpha \mathcal{H}(\widehat{\mu}^{s_i', \pi_Z(s_i')}, q_\theta^{s_i, a_i})],
\tag{24}
$$

*Proof.* Firstly, given a fixed $p(x)$ we know that minimizing $D_{\mathrm{KL}}(p, q_\theta)$ is equivalent to minimizing $\mathcal{H}(p, q)$ by following

$$
\begin{aligned}
D_{\mathrm{KL}}(p, q_\theta) &= \sum_{i=1}^N \int_{l_{i-1}}^{l_i} p_i(x)/\Delta \log \frac{p^i(x)/\Delta}{q_\theta^i/\Delta} \, \mathrm{d}x \\
&= -\sum_{i=1}^N \int_{l_{i-1}}^{l_i} p_i(x)/\Delta \log q_\theta^i/\Delta \, \mathrm{d}x - \left(\sum_{i=1}^N \int_{l_{i-1}}^{l_i} p_i(x)/\Delta \log p^i(x)/\Delta \, \mathrm{d}x\right) \\
&= \mathcal{H}(p, q_\theta) - \mathcal{H}(p) \\
&\propto \mathcal{H}(p, q_\theta)
\end{aligned}
\tag{25}
$$

where $p = \sum_{i=1}^N p_i(x) \mathbb{1}(x \in \Delta^i)/\Delta$ and $q_\theta = \sum_{i=1}^N q_i/\Delta$. Based on $\mathcal{H}(p, q_\theta)$, we use $p^{s_i', \pi_Z(s_i')}(x)$ to denote the target probability density function of the random variable $R(s_i, a_i) +$

$\gamma Z_{\theta^*}^k\left(s_i', \pi_Z(s_i')\right)$. Then, we can derive the objective function within each Neural FZI as

$$
\begin{aligned}
&\frac{1}{n}\sum_{i=1}^{n}\mathcal{H}(p^{s_i', \pi_Z(s_i')}(x), q_\theta^{s_i, a_i})\\
&= \frac{1}{n}\sum_{i=1}^{n}\left((1-\epsilon)\mathcal{H}(\mathbb{1}(x\in\Delta_E^i)/\Delta, q_\theta^{s_i, a_i}) + \epsilon\mathcal{H}(\widehat{\mu}^{s_i', \pi_Z(s_i')}, q_\theta^{s_i, a_i})\right)\\
&= \frac{1}{n}\sum_{i=1}^{n}\left(-(1-\epsilon)\sum_{j=1}^{N}\int_{l_{j-1}}^{l_j}\mathbb{1}(x\in\Delta_E^i)/\Delta\log q_\theta^{s_i, a_i}(\Delta_j)/\Delta dx - \epsilon\sum_{j=1}^{N}\int_{l_{j-1}}^{l_j}p_j^\mu/\Delta\log q_\theta^{s_i, a_i}(\Delta_j)/\Delta\right)\\
&= \frac{1}{n}\sum_{i=1}^{n}\frac{1}{\Delta}\left((1-\epsilon)(-\log q_\theta^{s_i, a_i}(\Delta_E^i)/\Delta) - \epsilon\sum_{j=1}^{N}p_j^\mu\log q_\theta^{s_i, a_i}(\Delta_j)/\Delta\right)\\
&\propto \frac{1}{n}\sum_{i=1}^{n}\left((1-\epsilon)(-\log q_\theta^{s_i, a_i}(\Delta_E^i)) + \epsilon\mathcal{H}(\widehat{\mu}^{s_i', \pi_Z(s_i')}, q_\theta^{s_i, a_i})\right)\\
&\propto \frac{1}{n}\sum_{i=1}^{n}\left(-\log q_\theta^{s_i, a_i}(\Delta_E^i) + \alpha\mathcal{H}(\widehat{\mu}^{s_i', \pi_Z(s_i')}, q_\theta^{s_i, a_i})\right), \text{ where } \alpha = \frac{\epsilon}{1-\epsilon} > 0
\end{aligned}
$$
(26)

where recall that $\widehat{\mu}^{s_i', \pi_Z(s_i')} = \sum_{i=1}^{N}p_i^\mu(x)\mathbb{1}(x\in\Delta_i)/\Delta = \sum_{i=1}^{N}p_i^\mu/\Delta$ for conciseness and denote $q_\theta^{s_i, a_i} = \sum_{j=1}^{N}q_\theta^{s_i, a_i}(\Delta_j)/\Delta$. The cross-entropy $\mathcal{H}(\widehat{\mu}^{s_i', \pi_Z(s_i')}, q_\theta^{s_i, a_i})$ is based on the discrete distribution when $i = 1, ..., N$. $\Delta_E^i$ represent the interval that $\mathbb{E}\left[R(s_i, a_i) + \gamma Z_{\theta^*}^k\left(s_i', \pi_Z(s_i')\right)\right]$ falls into, i.e., $\mathbb{E}\left[R(s_i, a_i) + \gamma Z_{\theta^*}^k\left(s_i', \pi_Z(s_i')\right)\right] \in \Delta_E^i$. $\qquad\square$

## G    Proof of Proposition 3

**Proposition 3** (Equivalence between **the term (a)** in Decomposed Neural FZI and Neural FQI) In Eq. 5 of Neural FZI, if the function class $\{Z_\theta : \theta \in \Theta\}$ is sufficiently large such that it contains the target $\{Y_i\}_{i=1}^n$. As $\Delta \to 0$, for $\forall k$, minimizing **the term (a)** in Eq. 5 implies

$$
P(Z_\theta^{k+1}(s, a) = \mathcal{T}^{\text{opt}}Q_{\theta^*}^k(s, a)) = 1, \quad \text{and} \quad \int_{-\infty}^{+\infty}\left|F_{q_\theta}(x) - F_{\delta_{\mathcal{T}^{\text{opt}}Q_{\theta^*}^k(s, a)}}(x)\right|dx = o(\Delta),
$$
(27)

where $\delta_{\mathcal{T}^{\text{opt}}Q_{\theta^*}^k(s, a)}$ is the delta function defined on $\mathcal{T}^{\text{opt}}Q_{\theta^*}^k(s, a)$.

*Proof.* Firstly, we define the distributional Bellman optimality operator $\mathfrak{T}^{\text{opt}}$ as follows:

$$
\mathfrak{T}^{\text{opt}}Z(s, a) \overset{D}{=} R(s, a) + \gamma Z\left(S', a^*\right),
$$
(28)

where $S' \sim P(\cdot \mid s, a)$ and $a^* = \underset{a'}{\arg\max}\,\mathbb{E}\left[Z\left(S', a'\right)\right]$. If $\{Z_\theta : \theta \in \Theta\}$ is sufficiently large enough such that it contains $\mathfrak{T}^{\text{opt}}Z_{\theta^*}\left(\{Y_i\}_{i=1}^n\right)$, then optimizing Neural FZI in Eq. 2 leads to $Z_\theta^{k+1} = \mathfrak{T}^{\text{opt}}Z_{\theta^*}$.

We apply the action-value density function decomposition on the target histogram function $\widehat{p}^{s, a}(x)$. Consider the parameterized histogram density function $h_\theta$ and denote $h_\theta^E/\Delta$ as the bin height in the bin $\Delta_E$, under the KL divergence between the first histogram function $\mathbb{1}(x\in\Delta_E)$ with $h_\theta(x)$, the objective function is simplified as

$$
D_{\text{KL}}(\mathbb{1}(x\in\Delta_E)/\Delta, h_\theta(x)) = -\int_{x\in\Delta_E}\frac{1}{\Delta}\log\frac{\frac{h_\theta^E}{\Delta}}{\frac{1}{\Delta}}dx = -\log h_\theta^E
$$
(29)

Since $\{Z_\theta : \theta \in \Theta\}$ is sufficiently large enough that can represent the pdf of $\{Y_i\}_{i=1}^n$, it also implies that $\{Z_\theta : \theta \in \Theta\}$ can represent the term (a) part in its pdf via the return density decomposition. The KL minimizer would be $\widehat{h}_\theta = \mathbb{1}(x\in\Delta_E)/\Delta$ in expectation. Then,

$\lim_{\Delta \to 0} \arg\min_{h_\theta} D_{\text{KL}}(\mathbb{1}(x \in \Delta_E)/\Delta, h_\theta(x)) = \delta_{\mathbb{E}[Z^{\text{target}}(s,a)]}$, where $\delta_{\mathbb{E}[Z^{\text{target}}(s,a)]}$ is a Dirac Delta function centered at $\mathbb{E}\left[Z^{\text{target}}(s,a)\right]$ and can be viewed as a generalized probability density function. The limit behavior from a histogram function $\widehat{p}$ to a continuous one for $Z^{\text{target}}$ is guaranteed by Theorem 1, and this also applies from $h_\theta$ to $Z_\theta$. In Neural FZI, we have $Z^{\text{target}} = \mathfrak{T}^{\text{opt}} Z_{\theta^*}$. According to the definition of the Dirac function, as $\Delta \to 0$, we attain

$$P(Z_\theta^{k+1}(s,a) = \mathbb{E}\left[\mathfrak{T}^{\text{opt}} Z_{\theta^*}^k(s,a)\right]) = 1 \tag{30}$$

Due to the linearity of expectation analyzed in Lemma 4 of (Bellemare et al., 2017a), we have

$$\mathbb{E}\left[\mathfrak{T}^{\text{opt}} Z_{\theta^*}^k(s,a)\right] = \mathfrak{T}^{\text{opt}} \mathbb{E}\left[Z_{\theta^*}^k(s,a)\right] = \mathcal{T}^{\text{opt}} Q_{\theta^*}^k(s,a) \tag{31}$$

Finally, we obtain:

$$P(Z_\theta^{k+1}(s,a) = \mathcal{T}^{\text{opt}} Q_{\theta^*}^k(s,a)) = 1 \quad \text{as } \Delta \to 0 \tag{32}$$

In order to characterize how the difference varies when $\Delta \to 0$, we further define $\Delta_E = [l_e, l_{e+1})$ and we have:

$$
\begin{aligned}
\int_{-\infty}^{+\infty} \left| F_{q_\theta}(x) - F_{\delta_{\mathcal{T}^{\text{opt}} Q_{\theta^*}^k(s,a)}}(x) \right| dx &= \frac{1}{2\Delta} \left( \left(\mathcal{T}^{\text{opt}} Q_{\theta^*}^k(s,a) - l_e\right)^2 + \left(l_{e+1} - \mathcal{T}^{\text{opt}} Q_{\theta^*}^k(s,a)\right)^2 \right) \\
&= \frac{1}{2\Delta}(a^2 + (\Delta - a)^2) \\
&\leq \Delta/2 \\
&= o(\Delta),
\end{aligned}
\tag{33}
$$

where $\mathcal{T}^{\text{opt}} Q_{\theta^*}^k(s,a) = \mathbb{E}\left[\mathfrak{T}^{\text{opt}} Z_{\theta^*}^k(s,a)\right] \in \Delta_E$ and we denote $a = \mathcal{T}^{\text{opt}} Q_{\theta^*}^k(s,a) - l_e$. The first equality holds as $q_\theta(x)$, the KL minimizer while minimizing the term (a), would follows a uniform distribution on $\Delta_E$, i.e., $\widehat{q}_\theta = \mathbb{1}(x \in \Delta_E)/\Delta$. Thus, the integral of LHS would be the area of two centralized triangles according. The inequality is because the maximizer is obtained when $a = \Delta$ or 0.

$\square$

# H CONVERGENCE PROOF OF DERPI IN THEOREM 2

## H.1 PROOF OF DISTRIBUTION-ENTROPY-REGULARIZED POLICY EVALUATION IN LEMMA 1

**Lemma 1**(Distribution-Entropy-Regularized Policy Evaluation) Consider the distribution-entropy-regularized Bellman operator $\mathcal{T}_d^\pi$ in Eq. 8 and assume $\mathcal{H}(\mu^{s_t,a_t}, q_\theta^{s_t,a_t}) \leq M$ for all $(s_t, a_t) \in \mathcal{S} \times \mathcal{A}$, where $M$ is a constant. Define $Q^{k+1} = \mathcal{T}_d^\pi Q^k$, then $Q^{k+1}$ will converge to a *corrected* Q-value of $\pi$ as $k \to \infty$ with the new objective function $J'(\pi)$ defined as

$$J'(\pi) = \sum_{t=0}^T \mathbb{E}_{(s_t,a_t) \sim \rho_\pi} \left[ r(s_t, a_t) + \gamma f(\mathcal{H}(\mu^{s_t,a_t}, q_\theta^{s_t,a_t})) \right]. \tag{34}$$

*Proof.* Firstly, we plug in $V(s_{t+1})$ into RHS of the iteration in Eq. 8, then we obtain

$$
\begin{aligned}
&\mathcal{T}_d^\pi Q(s_t, a_t) \\
&= r(s_t, a_t) + \gamma \mathbb{E}_{s_{t+1} \sim P(\cdot | s_t, a_t)} [V(s_{t+1})] \\
&= r(s_t, a_t) + \gamma f(\mathcal{H}(\mu^{s_t,a_t}, q_\theta^{s_t,a_t})) + \gamma \mathbb{E}_{(s_{t+1},a_{t+1}) \sim \rho^\pi} [Q(s_{t+1}, a_{t+1})] \\
&\triangleq r_\pi(s_t, a_t) + \gamma \mathbb{E}_{(s_{t+1},a_{t+1}) \sim \rho^\pi} [Q(s_{t+1}, a_{t+1})],
\end{aligned}
\tag{35}
$$

where $r_\pi(s_t, a_t) \triangleq r(s_t, a_t) + \gamma f(\mathcal{H}(\mu^{s_t,a_t}, q_\theta^{s_t,a_t}))$ is the entropy augmented reward we redefine. Applying the standard convergence results for policy evaluation (Sutton & Barto, 2018), we can attain that this Bellman updating under $\mathcal{T}_d^\pi$ is convergent under the assumption of $|\mathcal{A}| < \infty$ and bounded entropy augmented rewards $r_\pi$. $\square$

## H.2   POLICY IMPROVEMENT WITH PROOF

**Lemma 2.** *(Distribution-Entropy-Regularized Policy Improvement) Let $\pi \in \Pi$ and a new policy $\pi_{new}$ be updated via the policy improvement step in the policy optimization. Then $Q^{\pi_{new}}(s_t, a_t) \geq Q^{\pi_{old}}(s_t, a_t)$ for all $(s_t, a_t) \in \mathcal{S} \times \mathcal{A}$ with $|\mathcal{A}| \leq \infty$.*

*Proof.* The policy improvement in Lemma 2 implies that $\mathbb{E}_{a_t \sim \pi_{\text{new}}}\left[Q^{\pi_{\text{old}}}(s_t, a_t)\right] \geq \mathbb{E}_{a_t \sim \pi_{\text{old}}}\left[Q^{\pi_{\text{old}}}(s_t, a_t)\right]$, we consider the Bellman equation via the distribution-entropy-regularized Bellman operator $\mathcal{T}_{sd}^{\pi}$:

$$
\begin{aligned}
Q^{\pi_{\text{old}}}(s_t, a_t) &\triangleq r(s_t, a_t) + \gamma \mathbb{E}_{s_{t+1} \sim \rho}\left[V^{\pi_{\text{old}}}(s_{t+1})\right] \\
&= r(s_t, a_t) + \gamma f(\mathcal{H}(\mu^{s_t, a_t}, q_\theta^{s_t, a_t})) + \gamma \mathbb{E}_{(s_{t+1}, a_{t+1}) \sim \rho^{\pi_{\text{old}}}}\left[Q^{\pi_{\text{old}}}(s_{t+1}, a_{t+1})\right] \\
&\leq r(s_t, a_t) + \gamma f(\mathcal{H}(\mu^{s_t, a_t}, q_\theta^{s_t, a_t})) + \gamma \mathbb{E}_{(s_{t+1}, a_{t+1}) \sim \rho^{\pi_{\text{new}}}}\left[Q^{\pi_{\text{old}}}(s_{t+1}, a_{t+1})\right] \\
&= r_{\pi_{\text{new}}}(s_t, a_t) + \gamma \mathbb{E}_{(s_{t+1}, a_{t+1}) \sim \rho^{\pi_{\text{new}}}}\left[Q^{\pi_{\text{old}}}(s_{t+1}, a_{t+1})\right] \\
&\vdots \\
&\leq Q^{\pi_{\text{new}}}(s_{t+1}, a_{t+1}),
\end{aligned}
\tag{36}
$$

where we have repeated expanded $Q^{\pi_{\text{old}}}$ on the RHS by applying the distribution-entropy-regularized distributional Bellman operator. Convergence to $Q^{\pi_{\text{new}}}$ follows from Lemma 1. □

## H.3   PROOF OF DERPI IN THEOREM 2

**Theorem 2** (Distribution-Entropy-Regularized Policy Iteration) Assume $\mathcal{H}(\mu^{s_t, a_t}, q_\theta^{s_t, a_t}) \leq M$ for all $(s_t, a_t) \in \mathcal{S} \times \mathcal{A}$, where $M$ is a constant. Repeatedly applying distribution-entropy-regularized policy evaluation in Eq. 8 and the policy improvement, the policy converges to an optimal policy $\pi^*$ such that $Q^{\pi^*}(s_t, a_t) \geq Q^{\pi}(s_t, a_t)$ for all $\pi \in \Pi$.

*Proof.* The proof is similar to soft policy iteration (Haarnoja et al., 2018). For completeness, we provide the proof here. By Lemma 2, as the number of iteration increases, the sequence $Q^{\pi_i}$ at $i$-th iteration is monotonically increasing. Since we assume the risk-sensitive entropy is bounded by $M$, the $Q^{\pi}$ is thus bounded as the rewards are bounded. Hence, the sequence will converge to some $\pi^*$. Further, we prove that $\pi^*$ is in fact optimal. At the convergence point, for all $\pi \in \Pi$, it must be case that:

$$
\mathbb{E}_{a_t \sim \pi^*}\left[Q^{\pi_{\text{old}}}(s_t, a_t)\right] \geq \mathbb{E}_{a_t \sim \pi}\left[Q^{\pi_{\text{old}}}(s_t, a_t)\right].
$$

According to the proof in Lemma 2, we can attain $Q^{\pi^*}(s_t, a_t) > Q^{\pi}(s_t, a_t)$ for $(s_t, a_t)$. That is to say, the "corrected" value function of any other policy in $\Pi$ is lower than the converged policy, indicating that $\pi^*$ is optimal. □

## I   PROOF OF INTERPOLATION FORM OF $\hat{J}_q(\theta)$

$$
\begin{aligned}
\hat{J}_q(\theta) &= \mathbb{E}_{s,a}\left[\left(\mathcal{T}_d^{\pi} Q_{\theta^*}(s, a) - Q_\theta(s, a)\right)^2\right] \\
&= \mathbb{E}_{s,a}\left[\left(\mathcal{T}^{\pi} Q_{\theta^*}(s, a) - Q_\theta(s, a) + \gamma(\tau^{1/2} \mathcal{H}^{1/2}(\mu^{s,a}, q_\theta^{s,a})/\gamma)\right)^2\right] \\
&= \mathbb{E}_{s,a}\left[\left(\mathcal{T}^{\pi}\mathbb{E}\left[q_{\theta^*}(s, a)\right] - \mathbb{E}\left[q_\theta(s, a)\right]\right)^2\right] + \tau \mathbb{E}_{s,a}\left[\mathcal{H}(\mu^{s,a}, q_\theta^{s,a})\right] \\
&\quad + \mathbb{E}_{s,a}\left[\left(\mathcal{T}^{\pi}\mathbb{E}\left[q_{\theta^*}(s, a)\right] - \mathbb{E}\left[q_\theta(s, a)\right]\right) \mathcal{H}(\mu^{s,a}, q_\theta^{s,a})\right] \\
&= \mathbb{E}_{s,a}\left[\left(\mathcal{T}^{\pi}\mathbb{E}\left[q_{\theta^*}(s, a)\right] - \mathbb{E}\left[q_\theta(s, a)\right]\right)^2\right] + \tau \mathbb{E}_{s,a}\left[\mathcal{H}(\mu^{s,a}, q_\theta^{s,a})\right] \\
&\propto (1 - \lambda)\mathbb{E}_{s,a}\left[\left(\mathcal{T}^{\pi}\mathbb{E}\left[q_{\theta^*}(s, a)\right] - \mathbb{E}\left[q_\theta(s, a)\right]\right)^2\right] + \lambda \mathbb{E}_{s,a}\left[\mathcal{H}(\mu^{s,a}, q_\theta^{s,a})\right],
\end{aligned}
\tag{37}
$$

where the second equation is based on the definition of Distribution-Entropy-Regularized Bellman Operator $\mathcal{T}_d^{\pi}$ in Eq. 8 and let $f(\mathcal{H}) = (\tau \mathcal{H})^{1/2}/\gamma$. The last equation is based on Lemma 1 in (Shi et al., 2022), where we let $\varphi(S_t, A_t) = \mathcal{H}(\mu^{S_t, A_t}, q_\theta^{S_t, A_t})$, and thus we have $\mathbb{E}_{s,a}\left[(\mathcal{T}^{\pi}\mathbb{E}\left[q_{\theta^*}(s, a)\right] - \mathbb{E}\left[q_\theta(s, a)\right]) \mathcal{H}(\mu^{s,a}, q_\theta^{s,a})\right] = 0$. We set $\lambda = \frac{\tau}{1+\tau} \in [0, 1]$.

## J IMPLEMENTATION DETAILS

Table 1: Hyper-parameters Sheet.

| Hyperparameter | Value |
|---|---|
| *Shared* | |
| Policy network learning rate | 3e-4 |
| (Quantile) Value network learning rate | 3e-4 |
| Optimization | Adam |
| Discount factor | 0.99 |
| Target smoothing | 5e-3 |
| Batch size | 256 |
| Replay buffer size | 1e6 |
| Minimum steps before training | 1e4 |
| *DSAC with C51* | |
| Number of Atoms ($N$) | 51 |
| *DSAC with IQN* | |
| Number of quantile fractions ($N$) | 32 |
| Quantile fraction embedding size | 64 |
| Huber regression threshold | 1 |

| Hyperparameter | Temperature Parameter $\beta$ | Max episode lenght |
|---|---|---|
| Walker2d-v2 | 0.2 | 1000 |
| Swimmer-v2 | 0.2 | 1000 |
| Reacher-v2 | 0.2 | 1000 |
| Ant-v2 | 0.2 | 1000 |
| HalfCheetah-v2 | 0.2 | 1000 |
| Humanoid-v2 | 0.05 | 1000 |
| HumanoidStandup-v2 | 0.05 | 1000 |
| BipedalWalkerHardcore-v2 | 0.002 | 2000 |

Our implementation is directly adapted from the source code in (Ma et al., 2020).

For Distributional SAC with C51, we use 51 atoms similar to the C51 (Bellemare et al., 2017a). For distributional SAC with quantile regression, instead of using fixed quantiles in QR-DQN, we leverage the quantile fraction generation based on IQN (Dabney et al., 2018a) that uniformly samples quantile fractions in order to approximate the full quantile function. In particular, we fix the number of quantile fractions as $N$ and keep them in ascending order. Besides, we adapt the sampling as $\tau_0 = 0, \tau_i = \epsilon_i / \sum_{i=0}^{N-1}$, where $\epsilon_i \in U[0, 1], i = 1, ..., N$.

### J.1 HYPER-PARAMETERS AND NETWORK STRUCTURE.

We adopt the same hyper-parameters, which are listed in Table 1 and network structure as in the original distributional SAC paper (Ma et al., 2020).

## K DERAC ALGORITHM

We provide a detailed algorithm description of DERAC algorithm in Algorithm 1.

## L EXPERIMENTS: SENSITIVITY ANALYSIS OF DERAC

Figure 5 suggests that C51 with cross-entropy loss behaves similarly to the vanilla C51 equipped with KL divergence.

---

**Algorithm 1** Distribution-Entropy-Regularized Actor Critic (DERAC) Algorithm

---

1: Initialize two value networks $q_\theta$, $q_{\theta^*}$, and policy network $\pi_\phi$.
2: **for** each iteration **do**
3:     **for** each environment step **do**
4:         $a_t \sim \pi_\phi(a_t|s_t)$.
5:         $s_{t+1} \sim p(s_{t+1}|s_t, a_t)$.
6:         $\mathcal{D} \leftarrow \mathcal{D} \cup \{(s_t, a_t, r(s_t, a_t), s_{t+1})\}$
7:     **end for**
8:     **for** each gradient step **do**
9:         $\theta \leftarrow \theta - \lambda_q \nabla_\theta \hat{J}_q(\theta)$
10:        $\phi \leftarrow \phi + \lambda_\pi \nabla_\phi \hat{J}_\pi(\phi)$.
11:        $\theta^* \leftarrow \tau\theta + (1 - \tau)\theta^*$
12:     **end for**
13: **end for**

---

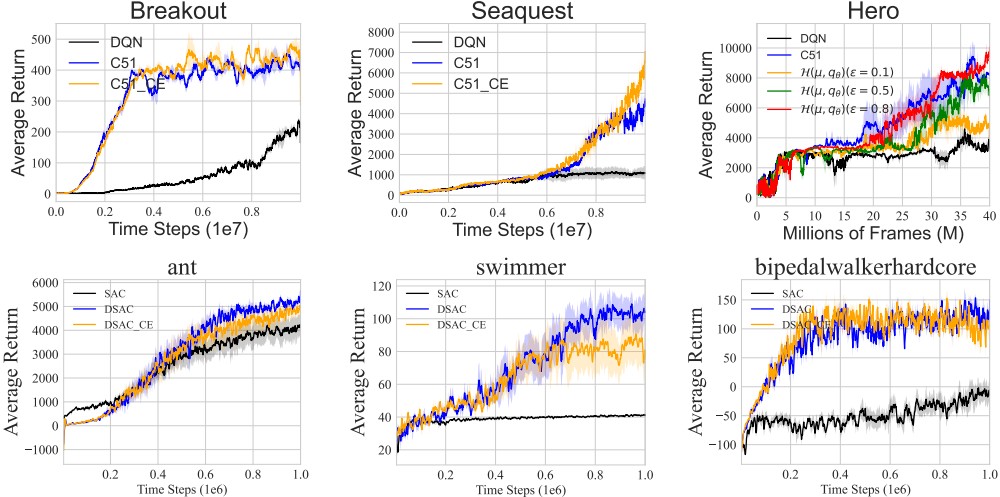

Figure 5: (**First row**) Learning curves of C51 under cross-entropy loss on Atari games over 3 seeds. (**Second row**) Learning curves of DSAC with C51 under cross-entropy loss on MuJoCo environments over 5 seeds.

Figure 6 shows that DERAC with different $\lambda$ in Eq. 11 may behave differently in the different environment. Learning curves of DERAC with an increasing $\lambda$ will tend to DSAC (C51), e.g., Bipedalwalkerhardcore, where DERAC with $\lambda = 1$ in the green line tends to DSAC (C51) in the blue line. However, DERAC with a small $\lambda$ is likely to outperform DSAC (C51) by only leveraging the expectation effect of return distribution, on Bipedalwalkerhardcore, where DERAC with $\lambda = 0, 0.5$ bypass DERAC with $\lambda = 1.0$.

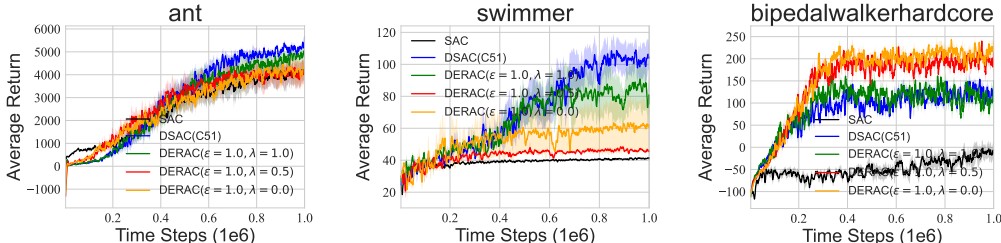

Figure 6: Learning curves of DERAC algorithms across different $\lambda$ on three MuJoCo environments over 5 seeds.

## M  EXPERIMENTS: MORE COMPLICATED ENVIRONMENTS

We further conduct experiments on more complicated environments, including Humanoid (state space as 375, action space as 16) and Walker2D (state space 17, action space 16) to show that DE-RAC is more likely to outperform on complicated environments. As shown in Figure 7, DERAC (red line) without entropy is competitive to other baselines, including AC (or SAC) and DAC (C51) and especially is superior to DAC (C51) on both complicated environments. In particular, on humanoid, DERAC performs better than both SAC and DAC (C51), and bypasses both AC and DAC (C51) on Walker2D, suggesting its potential on complicated environments.

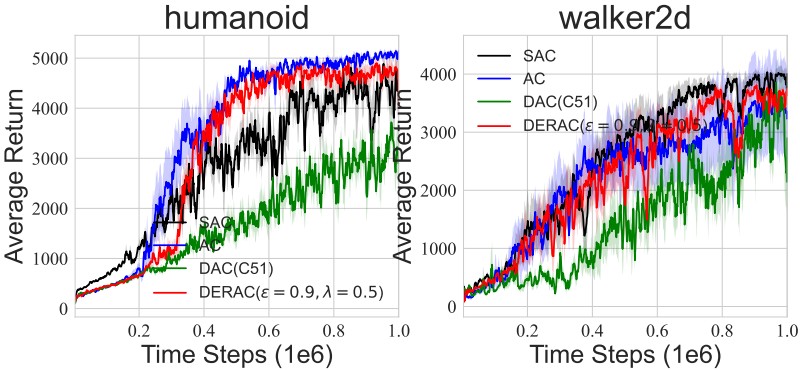

Figure 7: Learning curves of Distributional AC (C51) with the return distribution decomposition on complicated environments, including Humanoid and Walker2D. All results are averaged over 5 seeds. We denote AC as SAC without the leverage of entropy and DAC (C51) as DSAC (C15) without entropy.

## N  ENVIRONMENTS: MUTUAL IMPACTS ON DSAC (C51)

We presents results on 7 MuJoCo environments and omits Bipedalwalkerhardcore due to some engineering issue when the C51 algorithm interacts with the simulator. Figures 8 showcases that AC+RE (black) tends to perform better than AC (red) except on Humanoid and walker2d. However, when compared with AC+RE, AC+RE+VE (orange) may hurt the performance , e.g. on halfcheetah, ant and swimmer, while further boosts the performance on complicated environments, including

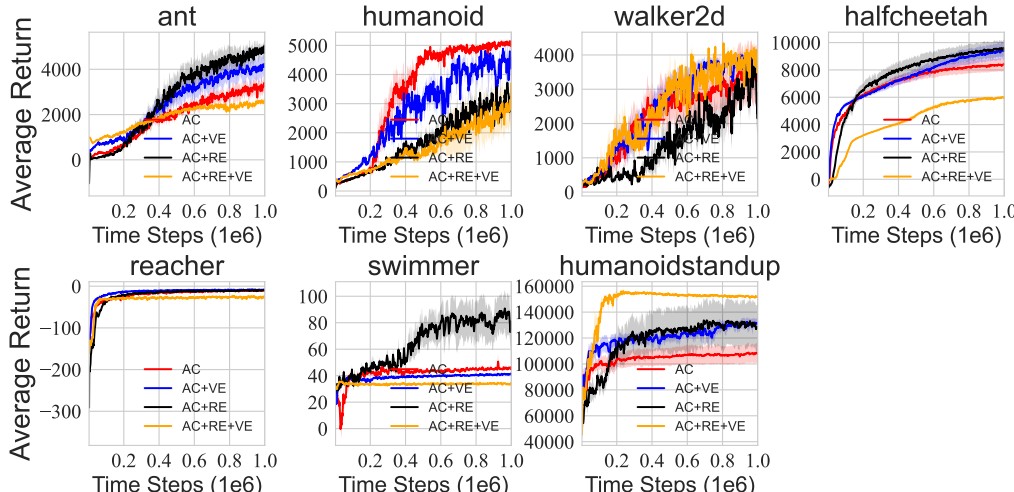

Figure 8: Learning curves of AC, AC+VE (SAC), AC+RE (DAC) and AC+RE+VE (DSAC) over 5 seeds with smooth size 5 across 7 MuJoCo environments where distributional RL part is based on **C51**.

humanoidstandup and walker2d. Similar situation is also applicable to AC+VE (blue). All of the conclusions made on DSAC (C51) is similar to DSAC (IQN).

