+1} = \operatorname*{argmin}_{Q_\theta} \frac{1}{n} \sum_{i=1}^{n} \left[ y_i - Q_\theta^k(s_i, a_i) \right]^2, \tag{1}$$

where the target $y_i = r(s_i, a_i) + \gamma \max_{a \in \mathcal{A}} Q_{\theta^*}^k(s_i', a)$ is fixed within every $T_{\text{target}}$ steps to update target network $Q_{\theta^*}$ by letting $\theta^* = \theta$. The experience buffer induces independent samples $\{(s_i, a_i, r_i, s_i')\}_{i \in [n]}$. In an ideal case when we neglect the non-convexity and TD approximation errors, we have $Q_\theta^{k+1} = \mathcal{T}^{\text{opt}} Q_{\theta^*}^k$, which is exactly the updating rule under Bellman optimality

operator (Fan et al., 2020). In the viewpoint of statistics, the optimization problem in Eq. 1 in each iteration is a standard supervised and neural network parameterized regression regarding $Q_\theta$.

**Distributional RL: Neural Fitted Z-Iteration (Neural FZI).** We interpret distributional RL as Neural FZI as it is by far closest to the practical algorithms, although our analysis is not intended for involving properties of neural networks. Analogous to Neural FQI, we can simplify value-based distributional RL algorithms parameterized by $Z_\theta$ into Neural FZI as

$$Z_\theta^{k+1} = \underset{Z_\theta}{\arg\min} \frac{1}{n} \sum_{i=1}^{n} d_p(Y_i, Z_\theta^k(s_i, a_i)), \tag{2}$$

where the target $Y_i = R(s_i, a_i) + \gamma Z_{\theta^*}^k(s_i', \pi_Z(s_i'))$ with the policy $\pi_Z$ following the greedy rule $\pi_Z(s_i') = \arg\max_{a'} \mathbb{E}\left[Z_{\theta^*}^k(s_i', a')\right]$ is fixed within every $T_{\text{target}}$ steps to update target network $Z_{\theta^*}$. $d_p$ is a divergence between two distributions. Notably, the choice of representation for $Z_\theta$ and the metric $d_p$ are pivotal for the empirical success of distributional RL algorithms (Sun et al., 2022b).

## 3.2 EQUIVALENT FORM OF DISTRIBUTIONAL RL: ENTROPY-REGULARIZED NEURAL FQI

**Return Density Function Decomposition.** To separate the impact of additional distribution information from the expectation of $Z^\pi$, we use a variant of *gross error model* from robust statistics (Huber, 2004), which was also similarly used to analyze Label Smoothing (Müller et al., 2019) and Knowledge Distillation (Hinton et al., 2015). Similar to the representation manner in CDRL (Dabney et al., 2018b) we utilize a *histgram function estimator* $\widehat{p}^{s,a}(x)$ with $N$ bins to approximate an arbitrary continuous action-value density function $p^{s,a}(x)$ given a state $s$ and action $a$. We leverage the continuous histogram estimator rather than the discrete categorical parameterization to allow richer analysis. Given a fixed set of supports $l_0 \leq l_1 \leq ... \leq l_N$ with the equal bin size as $\Delta$, $\Delta_i = [l_{i-1}, l_i), i = 1, ..., N-1$ with $\Delta_N = [l_{N-1}, l_N]$, the continuous histogram density function is $\widehat{p}^{s,a}(x) = \sum_{i=1}^{N} p_i \mathbb{1}(x \in \Delta_i)/\Delta$. Denote $\Delta_E$ as the interval that $\mathbb{E}[Z^\pi(s,a)]$ falls into, i.e., $\mathbb{E}[Z^\pi(s,a)] \in \Delta_E$. We have an action-state return density function decomposition over $\widehat{p}^{s,a}(x)$:

$$\widehat{p}^{s,a}(x) = (1 - \epsilon)\mathbb{1}(x \in \Delta_E)/\Delta + \epsilon\widehat{\mu}^{s,a} \tag{3}$$

where $\widehat{p}^{s,a}$ is decomposed into a single-bin histogram $\mathbb{1}(x \in \Delta_E)/\Delta$ and an induced one $\widehat{\mu}^{s,a}$ evaluated by $\widehat{\mu}^{s,a}(x) = \sum_{i=1}^{N} p_i^\mu \mathbb{1}(x \in \Delta_i)/\Delta$. We will show later in Proposition 3 that optimizing the first term in Neural FZI is equivalent to Neural FQI for expectation-based RL in the limiting case. Therefore, the second term $\mu^{s,a}$ is derived to characterize the impact of action-state return distribution *despite* its expectation $\mathbb{E}[Z^\pi(s,a)]$ on the performance of distributional RL algorithms. $\epsilon$ is a pre-specified hyper-parameter before the decomposition, which controls the proportion between $\mathbb{1}(x \in \Delta_E)/\Delta$ and $\widehat{\mu}^{s,a}(x)$. Before establishing the equivalence between distributional RL and a specific entropy-regularized Neural FQI, we begin by showing that $\widehat{\mu}(x)$ is a valid density function under certain $\epsilon$ in Proposition 1. The proof is provided in Appendix B.

**Proposition 1.** *(Decomposition Validity) Denote* $\widehat{p}^{s,a}(x \in \Delta_E) = p_E/\Delta$. *In Eq. 3,* $\widehat{\mu}(x) = \sum_{i=1}^{N} p_i^\mu \mathbb{1}(x \in \Delta_i)/\Delta$ *is a valid probability density function* $\iff \epsilon \geq 1 - p_E$.

Proposition 1 indicates that the return density function decomposition in Eq. 3 is strictly correct as the pre-specified hyper-parameter $\epsilon$ satisfies $\epsilon \geq 1 - p_E$. This implies that $\epsilon \to 0$ is not attainable, where distributional RL degrades to expectation-based RL (Proposition 3). As such, this decomposition exactly keeps the vanilla (categorical) distributional RL scenario without shifting problems.

**Histogram Function Parameterization Error: Uniform Convergence in Probability.** We begin by pointing out that the histogram density estimator is equivalent to the categorical parameterization with the proof given in Appendix C, although the former is a continuous estimator in contrast to the discrete nature of the latter. However, the previous discrete categorical parameterization error bound in (Rowland et al., 2018) (Proposition 3) is derived between the true return distribution and the limiting return distribution denoted as $\eta_C$ iteratively updated via the Bellman operator $\Pi_C \mathfrak{T}^\pi$ *in expectation*, without considering an asymptotic analysis when the number of sampled $\{s_i, a_i\}_{i=1}^n$ pairs goes to infinity. As a complementary result, we provide a uniform convergence rate for the histogram density estimator in the context of distributional RL. In this particular analysis within this subsection, we denote $\widehat{p}_C^{s,a}$ as the density function estimator for the true limiting return distribution $\eta_C$ via $\Pi_C \mathfrak{T}^\pi$ with its true density $p_C^{s,a}$. In Theorem 1, we show that the sample-based histogram

estimator $\widehat{p}_{\mathcal{C}}^{s,a}$ can approximate any arbitrary continuous limiting density function $p_{\mathcal{C}}^{s,a}$ under a mild condition. The proof is provided in Appendix D.

**Theorem 1.** *(Uniform Convergence Rate in Probability) Suppose $p_{\mathcal{C}}^{s,a}(x)$ is Lipschitz continuous and the support of a random variable is partitioned by N bins with bin size $\Delta$. Then*

$$\sup_x |\widehat{p}_{\mathcal{C}}^{s,a}(x) - p_{\mathcal{C}}^{s,a}(x)| = O(\Delta) + O_P\left(\sqrt{\frac{\log N}{n\Delta^2}}\right). \tag{4}$$

**Distributional RL: Entropy-regularized Neural FQI.** We apply the decomposition on the target action-value histogram density function and choose KL divergence as $d_p$ in Neural FZI. Let $\mathcal{H}(U, V)$ be the cross-entropy between two probability measures $U$ and $V$, i.e., $\mathcal{H}(U, V) = -\int_{x \in \mathcal{X}} U(x) \log V(x) \, \mathrm{d}x$. The target histogram density function $\widehat{p}^{s,a}$ is decomposed as $\widehat{p}^{s,a}(x) = (1 - \epsilon)\mathbb{1}(x \in \Delta_E)/\Delta + \epsilon\widehat{\mu}(x)$. We can derive the following entropy-regularized form for distributional RL in Proposition 2.

**Proposition 2.** *(Decomposed Neural FZI) Denote $q_\theta^{s,a}(x)$ as the histogram function of $Z_\theta^k(s, a)$ in Neural FZI. Based on Eq. 3 and KL divergence as $d_p$, Neural FZI in Eq. 2 is simplified as*

$$Z_\theta^{k+1} = \operatorname*{argmin}_{q_\theta} \frac{1}{n} \sum_{i=1}^n [\underbrace{-\log q_\theta^{s_i,a_i}(\Delta_E^i)}_{(a)} + \alpha\mathcal{H}(\widehat{\mu}^{s_i',\pi_Z(s_i')}, q_\theta^{s_i,a_i})], \tag{5}$$

where $\alpha = \varepsilon/(1 - \varepsilon) > 0$ and $\Delta_E^i$ represents the interval that $\mathbb{E}\left[Z^\pi(s_i', \pi_Z(s_i'))\right]$ falls into, i.e., $\mathbb{E}\left[Z^\pi(s_i', \pi_Z(s_i'))\right] \in \Delta_E^i$. $\widehat{\mu}^{s_i',\pi_Z(s_i')}$ is the resulting histogram density function in the next state-action pair $(s_i', \pi_Z(s_i'))$. The proof is given in Appendix F. In Proposition 3 with proof in Appendix G, we further show that minimizing the term (a) in Eq. 5 is equivalent to minimizing Neural FQI. For the uniformity of notation, we still use $s, a$ in the following analysis instead of $s_i, a_i$.

**Proposition 3.** *(Equivalence between **the term (a)** in Decomposed Neural FZI and Neural FQI) In Eq. 5 of Neural FZI, if the function class $\{Z_\theta : \theta \in \Theta\}$ is sufficiently large such that it contains the target $\{Y_i\}_{i=1}^n$. As $\Delta \to 0$, for $\forall k$, minimizing **the term (a)** in Eq. 5 implies*

$$P(Z_\theta^{k+1}(s, a) = \mathcal{T}^{opt}Q_{\theta*}^k(s, a)) = 1, \quad and \quad \int_{-\infty}^{+\infty} \left|F_{q_\theta}(x) - F_{\delta_{\mathbb{E}[Z^\pi(s,a)]}}(x)\right| dx = o(\Delta), \tag{6}$$

*where $\delta_{\mathbb{E}[Z^\pi(s,a)]}$ is the delta function defined on $\mathbb{E}\left[Z^\pi(s, a)\right]$.*

Given the fact that $Q_\theta^{k+1} = \mathcal{T}^{opt}Q_{\theta*}^k$ ideally in Neural FQI (Fan et al., 2020), we have $Z_\theta^{k+1} = \mathcal{T}^{opt}Q_{\theta*}^k$ with probability one under the assumption in Proposition 3. This indicates $Z_\theta^{k+1}$ may take other values instead of its expectation part as $\mathbb{E}\left[Z_\theta^{k+1}\right] = \mathcal{T}^{opt}Q_{\theta*}^k$, but the probability when these events for other values happen is 0.

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

 target action-state return distributions have a higher cross-entropy relative to the current density estimator $q_\theta^{s_t,a_t}$.*

For a comprehensive analysis and a detailed comparison with maximum entropy RL, we now shift our attention to the properties of our risk-sensitive entropy regularization in the framework of Actor Critic (AC). In Lemma 1, we first show that our Distribution-Entropy-Regularized Bellman operator $\mathcal{T}_d^\pi$ still inherits the convergence property in the policy evaluation phase with a cumulative augmented reward function as the new objective function.

**Lemma 1.** *(Distribution-Entropy-Regularized Policy Evaluation) Consider the distribution-entropy-regularized Bellman operator $\mathcal{T}_d^\pi$ in Eq. 8 and assume $\mathcal{H}(\mu^{s_t,a_t}, q_\theta^{s_t,a_t}) \leq M$ for all $(s_t, a_t) \in \mathcal{S} \times \mathcal{A}$, where $M$ is a constant. Define $Q^{k+1} = \mathcal{T}_d^\pi Q^k$, then $Q^{k+1}$ will converge to a corrected Q-value of $\pi$ as $k \to \infty$ with the new objective function $J'(\pi)$ defined as*

$$J'(\pi) = \sum_{t=0}^{T} \mathbb{E}_{(s_t,a_t)\sim\rho_\pi} \left[ r\left(s_t, a_t\right) + \gamma f(\mathcal{H}\left(\mu^{s_t,a_t}, q_\theta^{s_t,a_t}\right)) \right]. \tag{10}$$

In the policy improvement for distributional RL, we keep the vanilla updating rules according to $\pi_{\text{new}} = \arg\max_{\pi'\in\Pi} \mathbb{E}_{a_t\sim\pi'} \left[Q^{\pi_{\text{old}}}(s_t, a_t)\right]$. Next, we can immediately derive a new policy iteration algorithm, called *Distribution-Entropy-Regularized Policy Iteration (DERPI)* that alternates between the policy evaluation in Eq. 8 and the policy improvement. It will provably converge to the policy with the optimal risk-sensitive entropy among all policies in $\Pi$ as shown in Theorem 2.

**Theorem 2.** *(Distribution-Entropy-Regularized Policy Iteration) Assume $\mathcal{H}(\mu^{s_t,a_t}, q_\theta^{s_t,a_t}) \leq M$ for all $(s_t, a_t) \in \mathcal{S} \times \mathcal{A}$, where $M$ is a constant. Repeatedly applying distribution-entropy-regularized policy evaluation in Eq. 8 and the policy improvement, the policy converges to an optimal policy $\pi^*$ such that $Q^{\pi^*}(s_t, a_t) \geq Q^\pi(s_t, a_t)$ for all $\pi \in \Pi$.*

Please refer to Appendix H for the proof of Lemma 1 and Theorem 2. Theorem 2 indicates that if we incorporate the risk-sensitive entropy regularization into the policy gradient framework in Eq. 10, we can design a variant of "soft policy iteration" (Haarnoja et al., 2018) that can guarantee the convergence to an optimal policy. Based on all the analysis above, we provide a comprehensive comparison between the explicit vanilla entropy in maximum entropy RL and the implicit risk-sensitive entropy in distributional RL.

**Explicit vs Implicit Policy Optimization and Exploration.** By comparing $J(\pi)$ in Eq. 7 and $J'(\pi)$ in Eq. 10, the state-wise entropy $\mathcal{H}(\pi(\cdot|s_t))$ is explicitly maximized *w.r.t.* $\pi$ in maximum entropy RL for policies with a higher entropy in terms of diverse actions. In contrast, distributional RL implicitly maximizes the risk-sensitive entropy regularization *w.r.t.* $\pi$ via $a_t \sim \pi(\cdot|s_t)$. Concretely, the learned policy is encouraged to *visit state $s_t$ along with the policy-determined action pairs via $a_t \sim \pi(\cdot|s_t)$ in the future whose target action-state return distributions have a higher cross-entropy in terms of the current density estimator $q_\theta^{s_t,a_t}$.* In expectation-based RL, the learned $q_\theta^{s_t,a_t}$ is more likely to concentrate on the expectation of target return distribution, without the leverage of the full return distribution information. Thus, opti-

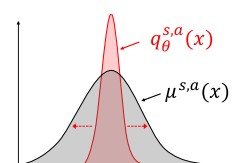

Figure 1: Risk-seeking scenario. $q_\theta^{s,a}$ is encouraged to disperse under the risk-sensitive entropy regularization in CDRL.

mizing the implicit regularization in distributional RL pushes $q_\theta^{s_t,a_t}$ to approach the target return distribution $\mu^{s_t,a_t}$ that tends to have a higher degree of dispersion, e.g., variance. As such, the implicit risk-sensitive entropy potentially promotes the **risk-sensitive exploration** to reduce the intrinsic uncertainty of the environment, as illustrated in Figure 1 for the risk-seeking scenario. It is still possible that $q_\theta^{s_t,a_t}$ has a higher variance already than $\mu^{s_t,a_t}$ in the learning process, corresponding to a risk-averse scenario. We argue that it highly depends on the environment to exactly determine which scenario happens for each state-action pair at a specific phase of learning.

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

(\mathbb{E}\left(Z^\pi(s,a)\right) - l_e\right)^2 + \left(l_{e+1} - \mathbb{E}\left(Z^\pi(s,a)\right)\right)^2 \right) \leq \Delta/2 \\
&= o(\Delta).
\end{aligned} \tag{33}
$$

$\square$

# H   CONVERGENCE PROOF OF DERPI IN THEOREM 2

## H.1   PROOF OF DISTRIBUTION-ENTROPY-REGULARIZED POLICY EVALUATION IN LEMMA 1

**Lemma 1**(Distribution-Entropy-Regularized Policy Evaluation) Consider the distribution-entropy-regularized Bellman operator $\mathcal{T}_d^\pi$ in Eq. 8 and assume $\mathcal{H}(\mu^{s_t,a_t}, q_\theta^{s_t,a_t}) \leq M$ for all $(s_t, a_t) \in \mathcal{S} \times \mathcal{A}$, where $M$ is a constant. Define $Q^{k+1} = \mathcal{T}_d^\pi Q^k$, then $Q^{k+1}$ will converge to a *corrected* Q-value of $\pi$ as $k \to \infty$ with the new objective function $J'(\pi)$ defined as

$$
J'(\pi) = \sum_{t=0}^T \mathbb{E}_{(s_t,a_t)\sim\rho_\pi}\left[r\left(s_t,a_t\right) + \gamma f(\mathcal{H}\left(\mu^{s_t,a_t}, q_\theta^{s_t,a_t}\right))\right]. \tag{34}
$$

*Proof.* Firstly, we plug in $V(s_{t+1})$ into RHS of the iteration in Eq. 8, then we obtain

$$
\begin{aligned}
&\mathcal{T}_d^\pi Q\left(s_t, a_t\right) \\
&= r\left(s_t, a_t\right) + \gamma \mathbb{E}_{s_{t+1}\sim P(\cdot|s_t,a_t)}\left[V\left(s_{t+1}\right)\right] \\
&= r\left(s_t, a_t\right) + \gamma f(\mathcal{H}\left(\mu^{s_t,a_t}, q_\theta^{s_t,a_t}\right)) + \gamma \mathbb{E}_{(s_{t+1},a_{t+1})\sim\rho^\pi}\left[Q\left(s_{t+1}, a_{t+1}\right)\right] \\
&\triangleq r_\pi\left(s_t, a_t\right) + \gamma \mathbb{E}_{(s_{t+1},a_{t+1})\sim\rho^\pi}\left[Q\left(s_{t+1}, a_{t+1}\right)\right],
\end{aligned} \tag{35}
$$

where $r_\pi\left(s_t, a_t\right) \triangleq r\left(s_t, a_t\right) + \gamma f(\mathcal{H}\left(\mu^{s_t,a_t}, q_\theta^{s_t,a_t}\right))$ is the entropy augmented reward we redefine. Applying the standard convergence results for policy evaluation (Sutton & Barto, 2018), we can attain that this Bellman updating under $\mathcal{T}_d^\pi$ is convergent under the assumption of $|\mathcal{A}| < \infty$ and bounded entropy augmented rewards $r_\pi$. $\square$

## H.2   POLICY IMPROVEMENT WITH PROOF

**Lemma 2.** *(Distribution-Entropy-Regularized Policy Improvement) Let $\pi \in \Pi$ and a new policy $\pi_{new}$ be updated via the policy improvement step in the policy optimization. Then $Q^{\pi_{new}}\left(s_t, a_t\right) \geq Q^{\pi_{old}}\left(s_t, a_t\right)$ for all $(s_t, a_t) \in \mathcal{S} \times \mathcal{A}$ with $|\mathcal{A}| \leq \infty$.*

*Proof.* The policy improvement in Lemma 2 implies that $\mathbb{E}_{a_t\sim\pi_{\mathrm{new}}}\left[Q^{\pi_{\mathrm{old}}}(s_t, a_t)\right] \geq \mathbb{E}_{a_t\sim\pi_{\mathrm{old}}}\left[Q^{\pi_{\mathrm{old}}}(s_t, a_t)\right]$, we consider the Bellman equation via the distribution-entropy-regularized Bellman operator $\mathcal{T}_{sd}^\pi$:

$$
\begin{aligned}
Q^{\pi_{\mathrm{old}}}\left(s_t, a_t\right) &\triangleq r\left(s_t, a_t\right) + \gamma \mathbb{E}_{s_{t+1}\sim\rho}\left[V^{\pi_{\mathrm{old}}}\left(s_{t+1}\right)\right] \\
&= r\left(s_t, a_t\right) + \gamma f(\mathcal{H}\left(\mu^{s_t,a_t}, q_\theta^{s_t,a_t}\right)) + \gamma \mathbb{E}_{(s_{t+1},a_{t+1})\sim\rho^{\pi_{\mathrm{old}}}}\left[Q^{\pi_{\mathrm{old}}}\left(s_{t+1}, a_{t+1}\right)\right] \\
&\leq r\left(s_t, a_t\right) + \gamma f(\mathcal{H}\left(\mu^{s_t,a_t}, q_\theta^{s_t,a_t}\right)) + \gamma \mathbb{E}_{(s_{t+1},a_{t+1})\sim\rho^{\pi_{\mathrm{new}}}}\left[Q^{\pi_{\mathrm{old}}}\left(s_{t+1}, a_{t+1}\right)\right] \\
&= r_{\pi^{\mathrm{new}}}\left(s_t, a_t\right) + \gamma \mathbb{E}_{(s_{t+1},a_{t+1})\sim\rho^{\pi_{\mathrm{new}}}}\left[Q^{\pi_{\mathrm{old}}}\left(s_{t+1}, a_{t+1}\right)\right] \\
&\vdots \\
&\leq Q^{\pi_{\mathrm{new}}}\left(s_{t+1}, a_{t+1}\right),