# OpenReview forum: "Interpreting Categorical Distributional Reinforcement Learning: An Implicit Risk-Sensitive Regularization Effect"
_ICLR.cc/2024/Conference — Submitted to ICLR 2024_

### Official Review · Reviewer_tqyH · 2023-10-20

**Soundness:** 3 good
**Presentation:** 2 fair
**Contribution:** 2 fair
**Rating:** 6
**Confidence:** 3

**Summary:**

The paper proposes a novel interpretation of the distributional Bellman operator based on entropy entropy regularization. Specifically, it represents the continuous return distribution by a histogram of fixed support with equal bin size. Then the return distribution can be decomposed into a single-bin histogram (i.e. the bin the expectation of the return distribution falls into) and a term characterizing the impact of the return distribution despite its expectation. The relative weights between the two terms are regulated by a pre-defined hyperparameter. The authors then move on to show that this decomposition of the return distribution allows the distributional Bellman operation (which they call Neural Fitted Z-Iteration) to be reframed as a cross entropy-regularized Neural FQI when the distribution divergence to minimize is chosen as the KL divergence. They've demonstrated convergence of the decomposed distributional Bellman operator when minimizing the KL divergence as a consequence of the equivalence between optimizing histogram function and categorical distribution. Finally they proposed a new actor-critic algorithm incorporating the cross-entropy regularization. Experiments on Atari and Mujoco demonstrated better overall performance and analyzed the interplay between two types of entropy regularization.

**Strengths:**

1. proposed and analyzed a novel angle to distributional RL
2. thorough experiments

**Weaknesses:**

1. I'm not so sure how enlighening the findings of the paper are. When you choose the KL divergence to minimize, equivalently you're maximizing cross-entropy. It doesn't come off as much of added value if you've shown the distributional Bellman operator minimizing KL is equivalent to a cross entropy regularized conventional Bellman operator when you approximate (I wouldn't say decompose, as the epsilon term renders your decomposition not exact) the return distribution as an expectation term and a categorical distribution.

2. (minor) insufficient rigourousness in basic concepts, e.g., in Introduction, Q-learning is a type of TD learning not a separate category. Also TD learning is not specific to expectation-based RL, most existing distributional RL including the referenced CDRL is TD learning (i.e. bootstrapping from other timesteps) w.r.t. return distributions. And CDRL doesn't seem an actor-critic to me as the policy is just argmax of expected atom values.

**Questions:**

1. how the epsilon in Eq. 3 is pre-specified? Should it not be dependent on the actual return distribution which cannot be known in advance? Am I correct in the understanding that you are actually not learning the full return distribution, but still its expectation albeit allowing some uncertainty around it?

2. if so, it seems to me you're matching the expaction whilst expanding it according to a pre-defined level of uncertainty retained from the target distribution. In this case, how would you ensure that the so expanded distribution still has the same expectation/the correct expectation?

3. if you are minimizing KL divergence, how would you ensure the return distribution model is and remains as a histogram which is count-based?

4. in Eq. 5, you're regularzing the cross-entropy between consecutive 'return distributions'. How is it connected to maximum-entropy RL which encourages 'policy' entropy or cross-entropy between the policy and a reference policy? There seems no bridge to me to be built between uncertainty in action selection and that in return estimate, the only connection perhaps being both can be approached by augmenting the reward function with the entropy term. In fact, I failed to understand the claim that your regularization is favouring target return distributions with large dispersion. It seems to me only to push the dispersion of the return distribution model to catch up with the dispersion of the target return distribution.

5. (minor) how and why a policy can be learned to reduce the intrinsic uncertainty of the environment, shouldn't it by definition be intrinsic and thus independent from the policy?

---

> ### Author Response · Authors · 2023-11-22
> **Author Response**
>
> Thank you for taking the time to review our paper. We appreciate your positive assessment and insightful feedback, and we would like to address the concerns you raised in your review.
>
> > how the epsilon in Eq. 3 is pre-specified? Should it not be dependent on the actual return distribution which cannot be known in advance? Am I correct in the understanding that you are actually not learning the full return distribution, but still its expectation albeit allowing some uncertainty around it?
>
> **A:** Eq.3 is to show that the return density can be decomposed via a pre-specified $\epsilon$, which controls the proportion of the expectation and the regularization part. Eq.3 does not imply that we are going to learn only based on the expectation, but to focus on the validity of the density decomposition. For example, in the sensitivity analysis in Figure 2, we are using the regularization term with different $\varepsilon$ to optimize the neural network in order to verify the impact of it.
>
> > if so, it seems to me you're matching the expaction whilst expanding it according to a pre-defined level of uncertainty retained from the target distribution. In this case, how would you ensure that the so expanded distribution still has the same expectation/the correct expectation?
>
> **A:** Yes. The expectation can be inherently maintained by simply taking the expectation over both sides in Eq. 3. Since the first part maintains the same/correct expectation with the weight $1-\epsilon$, the remaining term accordingly has the same expectation with the weight $\epsilon$.
>
>
> > if you are minimizing KL divergence, how would you ensure the return distribution model is and remains as a histogram which is count-based?
>
> **A:** The validity of the induced distribution is guaranteed in Proposition 1 for a certain range of $\epsilon$. The counted-based histogram is inherently maintained via the decomposition in Eq. 3.
>
>
> > in Eq. 5, you're regularzing the cross-entropy between consecutive 'return distributions'. How is it connected to maximum-entropy RL which encourages 'policy' entropy or cross-entropy between the policy and a reference policy? There seems no bridge to me to be built between uncertainty in action selection and that in return estimate, the only connection perhaps being both can be approached by augmenting the reward function with the entropy term. In fact, I failed to understand the claim that your regularization is favouring target return distributions with large dispersion. It seems to me only to push the dispersion of the return distribution model to catch up with the dispersion of the target return distribution.
>
>
> **A:** Let us look at Eq. 10 to compare with MaxEnt RL in Eq. 7. Since we know $a_i = \pi(\cdot | s_i)$, optimizing the regularization term implicitly updates the policy $\pi$. By contrast, MaxEnt RL explicitly update the policy via maximizing the entropy in Eq. 7
>
> In terms of Eq. 5, the regularization term in the value-based RL includes $q_\theta$, and optimizing Eq. 5 will directly affect the update of $q_\theta$. Note that $a_i =\arg\max_a \mathbb{E} [q_\theta^{s_i, a}]$, which therefore connects the return estimate with the action selection.
>
> We would like to clarify that we did not claim the regularization would certainly favor target return distribution with a large dispersion. **We state that it tends to do that as the original $q_\theta$ only captures the expectation and it tends to expand the distribution with a larger variance to match the distribution**. However, we agree your statement is safer and more accurate, which is that regularization in general helps to catch up with the dispersion of the target distribution. We have revised them appropriately in the updated paper.
>
> > (minor) how and why a policy can be learned to reduce the intrinsic uncertainty of the environment, shouldn't it by definition be intrinsic and thus independent from the policy?
>
>
> **A:**  The intrinsic uncertainty has been discussed in [1, 2], which is defined as the stochasticity of the environment, including the transition dynamics and stochastic reward function. Since distributional RL makes full use of the distribution information of return, it is deemed to reduce the intrinsic uncertainty of the environment for better performance.
>
>
> We thank the reviewer once again for the time and effort in reviewing our work! We are more than happy to answer any other questions you have.
>
> [1] Implicit Quantile Networks for Distributional Reinforcement Learning (ICML 2018)
>
> [2] Distributional Reinforcement Learning for Efficient Exploration (ICML 2019)

---

> > ### Comment · Reviewer_tqyH · 2023-11-22
> >
> > Thank you authors for your response.
> >
> > So if my understanding based on the paper and your responses is correct, the method does not actually learn the full return distribution but still just the expectation plus a predefined level of dispersion, with varying performance w.r.t. the choice of this level but no way of determining its optimal value in advance. I'm not so sure how valuable this design is.
> >
> > I understand that regularizing the return distribution affects the policy. What I failed to see is how encouraging the dispersion of the return distribution to match the dispersion of a target distribution is equivalent to encouraging the entropy of the policy as in MaxEnt RL, i.e. how uncertainty in return is translated to uncertainty in action.
> >
> > Lastly so you're capturing the intrinsic uncertainty, as opposed to reducing it, as it's out there objectively.
> >
> > Due to the above and also the weaknesses in my original review to which the authors didn't seem to object, I'll keep my score.

---

### Official Review · Reviewer_rBfi · 2023-10-30

**Soundness:** 2 fair
**Presentation:** 2 fair
**Contribution:** 2 fair
**Rating:** 3
**Confidence:** 3

**Summary:**

This paper studies why distributional reinforcement learning (RL) can achieve successful empirical results. They mainly investigate the Categorical Distributional RL (CDRL) and indicate its strength comes from risk-sensitive entropy regularization. They showed that this regularization serves as an augmented reward function, pushing the learned distribution toward the target one. Experiments highlight this regularization's importance in distributional RL.

**Strengths:**

This paper studies an important question (why distributional RL is successful), and the way that they consider the distributional RL (or specifically, the categorical RL) is interesting. The theoretical claims are supported by the experiments.

**Weaknesses:**

The paper is a bit hard to follow. Some of the theoretical derivation is not followed by intuitive explanation or proof sketch. For example, eq. (3) and the subsequent remarks confused me regarding what $\hat\mu$ and $p_i^\mu$ are. Does $p_i^\mu$ mean the coefficient of the corresponding bin? I also don't see what $\mu^{s,a}$ is two lines below. Moreover, I failed to understand eq. (6) and what Proposition 3 tries to convey, specifically regarding the integral on the right. Additionally, Theorem 2 asserts the convergence of the learned policy, yet it does not specify under what notion it converges, that is, it lacks clarity on the measure under which it converges.

I found most theoretical results rely on the assumption that $\mathcal{H}(\mu^{s,a},q_\theta^{s,a})$ is bounded by a constant for all $s,a$, which I think needs to be justified. I guess $\mu$ is the resulting histogram density, so the difference here might depend on the range of support and $N$, and thus the upper bound might actually look like the results in [1].

Overall, I think this paper may need some necessary polish to improve clarity before I can make an informed evaluation.



[1] Rowland, M., Bellemare, M., Dabney, W., Munos, R. and Teh, Y.W., 2018, March. An analysis of categorical distributional reinforcement learning. AISTATS, 2018.

**Questions:**

The formulation of neural FZI is similar to [2]. The latter focuses on maximizing the log-likelihood, which is equivalent to minimizing the KL divergence or cross-entropy. The authors also proposed to specify the $d_p$ in eq.(2) to KL divergence, so the formulation looks almost same. Hence, I am wondering about the potential connection here.

[2] Wu, R., Uehara, M. and Sun, W. Distributional Offline Policy Evaluation with Predictive Error Guarantees. ICML, 2023.

---

> ### Author Response · Authors · 2023-11-22
> **Author Response**
>
> Thank you for taking the time to review our paper. We appreciate your comments and we have revised the paper accordingly. Here we would like to address the concerns you raised in your review.
>
> > Eq. (3) and the subsequent remarks confused me regarding what $\widehat{\mu}$ is? Does mean the coefficient of the corresponding bin?
>
> **A:** We apologize for this confusion and we thus have provided the detailed definition of $\widehat{\mu}$  in the revised version. $p^\mu_{i}$  is indeed the coefficient of the corresponding bin with the bin height $p^\mu_{i}/\Delta$.
>
>
> >Moreover, I failed to understand eq. (6) and what Proposition 3 tries to convey, specifically regarding the integral on the right.
>
> **A:** Generally speaking, Eq. 6 and Proposition 3 try to convey that **minimizing the term (a) in the decomposed Neural FZI is equivalent to Neural FQI in the limit case**, allowing us to use the regularization term to interpret the benefits of distributional RL over classical RL analyzed later.
>
> The integral part additionally shows the **uniform convergence rate** of distribution difference in terms of $\Delta$, which turns out to be an order of $\mathcal{O}(\Delta)$.
>
> > Additionally, Theorem 2 asserts the convergence of the learned policy, yet it does not specify under what notion it converges, that is, it lacks clarity on the measure under which it converges.
>
> **A:** **The optimal policy is defined based on the optimal Q function**. In our proof, following SAC paper, we show the algorithm converges to the optimal Q function~(the maximum), and thus the corresponding policy/policies would be optimal. Please refer to  Appendix H.3 for more details.
>
> >I found most theoretical results rely on the assumption that the bounded entropy....
>
> **A:** The bounded assumption is natural and mild as the convergence of the Bellman operator typically requires **a bounded reward assumption**. Note that our cross-entropy term serves as the augmented rewards, for which it is natural to assume to be bounded to allow the convergence accordingly.
>
> >I think this paper may need some necessary polish to improve clarity before I can make an informed evaluation.
>
> **A:** Thanks for this suggestion and we have substantially improved the clarity based on all the reviewers' suggestions in the rebuttal phase. We appreciate an informed evaluation of our paper.
>
> >The formulation of neural FZI is similar to [2]. The latter focuses on maximizing the log-likelihood, which is equivalent to minimizing the KL divergence or cross-entropy. The authors also proposed to specify the divergence in eq.(2) to KL divergence, so the formulation looks almost the same. Hence, I am wondering about the potential connection here.
>
>
> **A:** [2] can be viewed as a **distributional version of Fitted Q evaluation in the offline setting**, whose target is to provide the **prediction error guarantee in the off-policy evaluation**. Based on the relationship of MLE they used and KL divergence, [2] can approximately serve as the prediction error guarantee for offline CDRL. However, our target is to interpret the benefits of distribution RL starting from CDRL, which is very different from [2].
>
> We would also like to clarify that [2] used **TV and Wasserstein distance instead of KL** in their theoretical analysis to derive the prediction error guarantee in an offline setting, which we focus on interpreting the KL-based practical CDRL algorithms in online cases.
>
> In summary, the main contribution and research target between [2] and us are different and based on different divergences, although the algorithm part shares some similarities with CDRL algorithm.
>
> We thank the reviewer once again for the time and effort in reviewing our work! We would greatly appreciate it if the reviewer could check our responses and let us know whether they address the raised concerns. We are more than happy to provide further clarification if you have any additional concerns. Should this rebuttal address your concerns, we would be grateful for a revised score.

---

### Official Review · Reviewer_73VF · 2023-10-30

**Soundness:** 2 fair
**Presentation:** 1 poor
**Contribution:** 3 good
**Rating:** 5
**Confidence:** 4

**Summary:**

In this work, the authors take a closer look at the categorical
distributional RL framework and study the properties of the
optimization problems therein. Inspired by their analysis, the paper
presents a novel framework (DERPI) for model-free RL similar to MaxEnt
RL, and an associatedc
actor-critic algorithm (DERAC) based on this framework. Optimization
in the DERPI framework is shown to be closely related to categorical
distributional RL, thereby establishing a connection between
categorical disatributional RL and MaxEnt RL.

**Strengths:**

This paper presents some interesting insights into the optimization of
return distributions in categorical distributional RL, and inspired by
these insights, proposes a new framework for exploration in RL. The
new framework is closely related to the highly successful MaxEnt RL,
but it performs reward shaping based on the mismatch between the
predicted return distribution and its target, which I believe is novel
and sensible. This framework also explicitly leverages the "auxiliary
tasks" quality of distributional RL for exploration, which is satisfying.

**Weaknesses:**

Many of the mathematical statements and objects are not defined
precisely -- this is a recurring issue throughout the text, making it
extremely difficult to follow. Please see the "Questions" section for
further details.

The histogram parameterization given in equation (3) needs more
clarification. Some terms are left undefined (e.g. $p_i^\mu$), making
it difficult to understand the model and the motivation.

As discussed more in "Questions", the return distribution
decomposition is fairly mysterious to me. It seems that the purpose is
to show that some component of CDRL is sort of 'mimicking' expected
value RL. However, this is already known, since it is known that the
categorical projection $\Pi_{\mathcal{C}}$ is mean-preserving. Thus,
I do not undertand the value of Proposition 3: we already know that
CDRL will learn the mean of the return distribution function.

Generally, I do not agree that the proposed algorithms are
risk-sensitive in any meaningful sense -- they simply have an
additional distribution-matching term in the reward signal. The
risk-seeking/risk-averse behavior is driven by a mismatch in return
distribution estimates and not a specified propensity for risk
(i.e. there is no way to control or predict how risk-averse the agent
will be).

The connection between DERPI and CDRL is not established clearly at
all, which is a shame, since I believe this is the most significant
finding. Particularly, I belive the content of Appendix I, notably
equation (37), should be clearly written in the main text and
discussed. As it stands, this part is easy to miss, and I believe it
is crucial for tying together the claims of the paper.

## Minor issues
On page 3 in the "Return Density Function Decomposition" paragraph,
there is a typo in "we utilize a histgram function estimator".

Proposition 1 would read better if the $\iff$ was substituted by the
text "if and only if".

**Questions:**

In the "Bellman Operators vs Distributional Bellman Operators"
section, it says tha random variable definition of the distributional
Bellman operator is less mathematically rigorous than the
return-distribution definition -- what do you mean by this? Both
definitions should be equivalent.

I doubt Theorem 1 is novel, it is essentially a concentration
inequality on histogram estimates. It is also not clear to me what is
meant by $O_P$ here. What is the difference between Theorem 1 and
e.g. the DKW inequality?

The object $\widehat{\mu}^{s,a}$ is not defined clearly at all.
Particularly, $p_i^\mu$ is not defined anywhere.
Is $\widehat{\mu}^{s,a}$ just the difference between the "true"
histogram of the return distribution function and the histogram with
all mass (times $1-\epsilon$) in $\Delta_E$?

Right after Proposition 1, what does "this decomposition exactly keeps
the vanilla (categorical) distributional RL scenario without shifting
problems" mean? What are shifting problems?

I don't agree that the second term of equation (5) is doing
"risk-sensitive entropy regularization". It is neither risk-seeking
nor risk-neutral (at least, not intuitively). Moreover, I wouldn't
classify this as a regularization term -- it is directly learning the
parameters of the model -- the only "regularization" here is the
explicit constraint of modeling categorical distributions. But
regardless of the parameterization of the return distribution
function, you can decompose the return distributions into a term
encapsulating the mean and the other encapsulating the rest of the
statistics. Is the idea that under the KL loss, the loss for the mean
component does not interact with the rest of the loss?


I don't understand the claim of equation (6). It says that with
probability 1, $Z_\theta^{k+1}=\mathcal{T}Q_{\theta^*}(s,a)$ -- how
can this be true? Isn't $Z$ supposed to represent the random return?
In this case, why should it be deterministic?

The proof of proposition 3 is sloppy. Firstly, equation (29) is incorrect -- the KL is not
proportional to the last line, it is missing an additive
$\Delta\log\Delta$ term I believe (though this does not change the
minimizer).
More importantly, the claim that the limiting optimizer as $\Delta\to
0$ is not proved correctly -- you proved that the limit of the
minimizer tends to the Dirac, but not that the minimizer of the limit
tends to the Dirac, as claimed. However, again I am not really
concerned about the correctness of the claim here.
Equation (30) is not justified to me at all -- the target should still
include the $\mu$ term, which seems to have disappeared. I believe the
confusion is coming from misleading notation. To my understanding, the
claim of Proposition 3 is that this is the behavior if you optimize
the random returns only w.r.t. term (a), but then that is not really
$Z$. So really, what this proposition is saying is that if the targets are
deterministic, they are at the expected return and CDRL will converge
to the Diracs at the expected return, but we already know this from
e.g. "An Analysis of Categorical Distributional Reinforcement
Learning" by Rowland et. al.

---

> ### Author Response · Authors · 2023-11-22
> **(1/2) Author Response**
>
> Thank you for taking the time to review our paper. We appreciate your comments and feedback, and we would like to address the concerns you raised in your review.
>
> >Q1: In the "Bellman Operators vs Distributional Bellman Operators" section, it says tha random variable definition ....
>
> **A:** This statement can refer to Chapter 4 of the book [1]. In particular, the random-variable operator may not be rigorous enough as it specifies a return distribution without identifying a mapping from the sample space to the real number, i.e., definite the probability measure.  However, the theory of contraction mappings needs a clear deﬁnition of the space on which an operator is deﬁned, i.e., the probability space of the return distribution defined in advance.
>
> We clearly state this difference in case any theory reviewers favor the more rigorous definition. However, we agree with you the difference is minor.
>
> >Q2: I doubt Theorem 1 is novel, it is essentially a concentration inequality on histogram estimates....
>
> **A:** The big O definition is a commonly used convergence process in probability theory, and here we provide the wiki reference: https://en.wikipedia.org/wiki/Big_O_in_probability_notation.  In particular, it characterizes a stochastic boundedness in the convergence in probability.
>
> The main similarity is both Theorem 1 and DKW are derived by leveraging the commonly used concentration inequality, and thus some parts share a similar form. However, the main difference is that DKW focuses on the (sample-based) empirical distribution, while Theorem 1 provides results about the histogram-based estimator in the context of CDRL.
>
> >Q3: The objection $\widehat{\mu}^{s, a}$ is not defined clearly ....
>
> **A:** Thank you for pointing out this clarity issue and we have fixed it according. It is true that $\widehat{\mu}$ is the induced histogram estimator, which is used to characterize the regularization effect / the benefits of distributional RL regardless of the expectation of return distributions.
>
> >Q4: Right after Proposition 1, what does "this decomposition exactly keeps the vanilla (categorical) distributional RL...
>
> **A:** We have paraphrased this sentence as ''Under this valid return density decomposition condition, this return density decomposition approach precisely maintains the standard categorical distribution framework in distributional RL''. This statement is made to clarify the potential concerns that this decomposition may change the objective function of CDRL. However, as stated in the new sentence, the original CDRL framework is carefully maintained.
>
> >Q5: I don't agree that the second term of equation (5) is doing "risk-sensitive entropy regularization"....
>
> **A:** We would like to clarify this potential misunderstanding. According to the literature [2], **risk** refers to uncertainty over possible outcomes, and **risk-sensitive policies** are those that depend upon more than the mean of the outcomes. This implies that our risk-sensitive policies, which are indeed risk-neutral, typically refer to the capability to reduce the intrinsic uncertainty of the environment. We know distributional RL provides a framework for risk-sensitive RL, including the risk-seeking or risk-averse policies, and therefore, it can be achieved **by changing the convexity or concavity of the utility function [2] on the return distribution** in the interaction phase of the environment, i.e., the behavior policy side. For example, IQN [2] changes different weights to average the quantiles to select the actions interacting with the environment. This risk-seeking or risk-averse framework can be very naturally incorporated into our algorithm or Neural FZI, by changing the greedy rule $\pi_Z(s^\prime_i)= \arg\max_{a^\prime} \mathbb{E}\left[Z_{\theta^*}^{k}(s_i^\prime, a^\prime)\right]$ in Eq. 2 with some specified utility function, $\pi_Z(s^\prime_i)= \arg\max_{a^\prime} U(Z_{\theta^*}^{k}(s_i^\prime, a^\prime))$.
>
> Regarding the regularization mechanism in Eq. 5, note that $a_i$ is selected by the policy, which is also determined by $q_\theta$. Thus, minimizing the term (a) and the regularization term will optimize the policy via $q_\theta$.

---

> ### Author Response · Authors · 2023-11-22
> **(2/2) Author Response**
>
> >Q6: I don't understand the claim of equation (6). It says that with probability.....
>
> **A:** $Z^{k+1}$ is a random variable and the convergence in probability 1 indicates that $Z^{k+1}$ would equal to a constant $\mathcal{T} Q_{\theta^*}$. The $\mathcal{T} Q_{\theta^*}$ is a constant in each Neural FZI as we use the target network trick, where the target is fixed in each iteration phase of Neural FZI.
>
> >Q7: The proof of proposition 3 is sloppy. Firstly, equation (29) is incorrect.....
>
> **A:** Thanks for pointing out some inaccurate parts in our proof and we have revised this proof accordingly, also including the limit of the minimizer issue in the Appendix.
>
> It is correct that proposition 3 is about optimizing the term (a) and thus the target distribution does not include the $\widehat{\mu}$ term. Regarding the relationship between [3], An Analysis of Categorical Distributional Reinforcement Learning" by Rowland et. al, note that [3] and other related works showed that most distributional RL algorithms can enjoy the mean preserving properties, but **our focus is to interpret the extra benefits of distributional RL over only this expectation part**. To this end, it is necessary to connect the term (a) with the optimization in terms of the expectation part, thus **allowing us to attribute the benefits to the regularization**. As far as we are concerned, proposition 3 is a crucial step towards interpreting the regularization effect. The results in proposition 3 actually coincide with the mean-preservation properties in distributional RL, which can help to substantiate the technical soundness of our decomposition.
>
> We thank the reviewer once again for the time and effort in reviewing our work! We would greatly appreciate it if the reviewer could check our responses and let us know whether they address the raised concerns. We are more than happy to provide further clarification if you have any additional concerns. Should this rebuttal address your concerns, we would be grateful for a revised score.
>
>
> [1] Bellmare et al. Distributional Reinforcement Learning (MIT press)
>
> [2] Dabney et al. Implicit quantile network for distributional reinforcement learning (ICML 2018)
>
> [3] Rowland et. al. An Analysis of Categorical Distributional Reinforcement Learning" (AISTATS 2018)

---

> > ### Comment · Reviewer_73VF · 2023-11-22
> >
> > Thanks to the authors for the responses. While this cleared up some of my questions, a few still remain.
> >
> > > ... risk refers to uncertainty over possible outcomes, and risk-sensitive policies are those that depend upon more than the mean of the outcomes. This implies that our risk-sensitive policies, which are indeed risk-neutral, typically refer to the capability to reduce the intrinsic uncertainty of the environment.
> >
> > I still think it's very misleading to call your algorithm "risk-sensitive". Basically every principled RL algorithm tries to reduce intrinsic uncertainty (this is the exploration problem), but they are generally not considered risk-sensitive. Moreover, it is not clear that this is even the phenomenon that your approach deals with -- the return distribution models aleatoric uncertainty, so the "risk-sensitive regularization" should not necessarily have the effect of reducing intrinsic uncertainty.
> >
> > > We know distributional RL provides a framework for risk-sensitive RL, including the risk-seeking or risk-averse policies, and therefore, it can be achieved by changing the convexity or concavity of the utility function [2] on the return distribution in the interaction phase of the environment, i.e., the behavior policy side.
> >
> > This is true, I guess, but it's a fairly weak argument in my opinion. Firstly, this is not what you do in the paper. This form of risk-sensitivity, like you say, can be employed in any distributional RL method, so it is not particular to your algorithm/analysis. Moreover, this is generally not even a principled method of performing risk-sensitive policy optimization: generally, the optimal risk-sensitive policy will not be {stationary, deterministic, Markov}.
> >
> > > $Z^{k+1}$ is a random variable and the convergence in probability 1 indicates that would equal to a constant $\mathcal{T}Q_{\theta^*}$
> >
> > I still don't understand this at all. I understand what the mathematical statement is saying, which to reiterate what you suggest, is that $Z^{k+1}\to\mathcal{T}Q_{\theta^*}$ in probability. But with stochastic dynamics, there is no reason that the return should be deterministic. My suspicion is that a key assumption is missing in the statement of this proposition, or at the very least, there is some miscommunication about what $Z^{k+1}$ represents here.

---

> > > ### Author Response · Authors · 2023-11-22
> > > **Author Further Response**
> > >
> > > Thanks for the reviewer's consistent dedication to reviewing our work.  We appreciate your comments and feedback and here is our explanation.
> > >
> > >
> > > > I still think it's very misleading to call your algorithm "risk-sensitive". Basically every principled RL algorithm tries to reduce intrinsic uncertainty (this is the exploration problem), but they are generally not considered risk-sensitive. Moreover, it is not clear that this is even the phenomenon that your approach deals with -- the return distribution models aleatoric uncertainty, so the "risk-sensitive regularization" should not necessarily have the effect of reducing intrinsic uncertainty.
> > >
> > > Firstly, based on the uncertainty quantification and distributional RL literature[1, 2], we would like to make sure both you and we also agree that the uncertainty in RL can be classified into **aleatoric (intrinsic) uncertainty**, which captures the stochasticity of the environment, and **epistemic (model or parameter) uncertainty**, which captures the stochasticity in the parameter estimate.
> > >
> > > To the best of our knowledge (which is also consistent with [1,2]), **distributional RL seeks to capture the intrinsic (aleatoric) uncertainty compared with classical RL**, as it uses the full return distribution in the algorithm instead of only its expectation. Since our decomposition maintains the CDRL framework and the equivalence established in Prop 3, the resulting regulation is supposed to be interpreted as the benefits to additionally capture the intrinsic uncertainty.
> > >
> > > However, we agree with your concern as the 'risk-sensitive' might be misleading in terms of risk-seeking or risk-averse policies used in risk-sensitive RL, while our work is instead focused on the part of dist RL that effectively captures the intrinsic uncertainty. We think **this issue mainly arises from our possibly direct usage of ``risk-sensitive policies'' from [2]**, although they use it to build the connection with risk-sensitive RL. We thus appreciate the reviewer raising this potentially misleading part and we will ensure to improve the clarity on this issue in the revised version.
> > >
> > >
> > > > This is true, I guess, but it's a fairly weak argument in my opinion. Firstly, this is not what you do in the paper. This form of risk-sensitivity, like you say, can be employed in any distributional RL method, so it is not particular to your algorithm/analysis. Moreover, this is generally not even a principled method of performing risk-sensitive policy optimization: generally, the optimal risk-sensitive policy will not be {stationary, deterministic, Markov}.
> > >
> > > Thanks for pointing out this issue again. We agree the risk-sensitive regularization we use might be used to connect risk-sensitive RL by readers, thus, being misleading in some sense (since we in fact focus on the intrinsic uncertainty). We would prefer to use **''(intrinsic) uncertainty regularization/exploration''** instead of ''risk-sensitive policy''. We will ensure you we will revise it in the final version and add a detailed discussion about the connection/difference with risk-sensitive RL as well. Thank again.
> > >
> > > >I still don't understand this at all. I understand what the mathematical statement is saying, which to reiterate what you suggest, is that in probability. But with stochastic dynamics, there is no reason that the return should be deterministic. My suspicion is that a key assumption is missing in the statement of this proposition, or at the very least, there is some miscommunication about what represents here.
> > >
> > > We guess you might miss that Proposition 3 is to connect the Neural FQI and **the first term (term (a))** in decomposed Neural FZI. In particular, if we use the term (a) to replace the (fixed) target return distribution based on our decomposition in each iteration of Neural FZI, $q_\theta$ tends to converge to the expectation of the target distribution, i.e., the constant $\mathcal{T} Q_{\theta^*}$ as $\Delta \rightarrow 0$. Also, note that $Z^{k+1}$ is the minimizer of $q_\theta$.
> > >
> > >
> > > We express our gratitude to the reviewer for their consistent dedication to reviewing our work. Should this rebuttal address your concerns, we would be grateful for a revised score. Of course, we remain at your disposal for any further clarifications.
> > >
> > > [1] Distributional Reinforcement Learning for Efficient Exploration (ICML 2019)
> > >
> > > [2] Dabney et al. Implicit quantile network for distributional reinforcement learning (ICML 2018)

---

### Official Review · Reviewer_FbE9 · 2023-11-01

**Soundness:** 1 poor
**Presentation:** 1 poor
**Contribution:** 1 poor
**Rating:** 3
**Confidence:** 4

**Summary:**

The paper establishes an equivalent representation of distributional RL (with a histogram representation of random variables) that seem to indicate that it takes the form of a regularized Q-iteration. This might also explain why categorical distributional RL might implicitly encourage exploration.

**Strengths:**

The paper investigates an important question: namely, how to explain the better performance of distributional RL algorithms. It articulates an innovative perspective on the question by decomposing the random variable representation as a mixture of a measure concentrated at the mean and a measure that embodies the spread.

**Weaknesses:**

The paper is highly obscure and filled with ambiguous statements and typos that prevented me from validating most of its claims. It also appear to support its theory using arguments that lack rigour or purely contradictory. I finally disagree with the claim on page 4 that distributional RL produces risk sensitive policies:

Examples of lack of rigour:
- the use of the concept of "ideal case" mentioned at the bottom of page 2 to support the idea that Neural FQI converges to optimal Q-function, which is reused in the proof of proposition 3 to assume that neural FZI's iterate recover the Bellman update.
- a mismatch of assumption between the statement of proposition 3 and its proof, where the former assumes that the $Y_i$'s are representable while the latter assumes that $E[Y_i]$ is.

Example of contradiction:
Proposition 3 is contradicting itself in equation (6). The left equation claims that the minimizer over $q_\theta$ converges to the k+1 iterate of the Q function in FQI while on the right claiming that it converges to the actual Q-value function.

Typos & inclarities include:
- p2: defining the support of \hat \eta first as $z_i$ then later by $l_i$
- Eq 1: $[y_i - Q_\theta(s_i,a_i)]^2$ instead of $[y_i - Q_\theta^k(s_i,a_i)]^2$
- Eq. 2: similar issue but for $Z_\theta^k(s_i,a_i)$
- p.3 "As such, this ... shifting problems" needs rewritting
- p.3 "the the"
- p.4 I suspect that in proposition 2 $\Delta_E^i$ represents the interval that $E[ R(s,a) + \gamma Z_k^\pi(s_i',\pi_Z(s_i') ]$ and that generally there is a confusion between this expression and $E[Z^\pi(s,a)]$.
- p.2  why define $R(s,a)$ with capital when it is a deterministic reward?
- p.3 in an expression like $Y_i = R(s_i,a_i)+\gamma Z_{\theta^*}^k(s_i',\pi_Z(s_i'))$ there needs to be a notation warning that lower cases variables are given.
- p4 the definition of $\hat \mu$ in proposition 2 is inaccurate as it seems to be the induced histogram from the decomposition of $Y_i$
- p4 "for $\forall$ k" should be "for all k"

**Questions:**

I do not have questions as I believe the paper needs a thorough rewriting and is currently unfit for publication.

---

> ### Author Response · Authors · 2023-11-22
> **Author Response**
>
> Thank you for taking the time to review our paper. We appreciate your comments and feedback, and we would like to address the concerns you raised in your review.
>
> >General Question: The paper is highly obscure and filled with ambiguous statements and typos that prevented me from validating most of its claims. It also appear to support its theory using arguments that lack rigour or purely contradictory. I finally disagree with the claim on page 4 that distributional RL produces risk sensitive policies:
>
> **A:** We apologize for the confusion generated by the clarity issue. Thus, we have improved our claim following your suggestions throughout the paper as appropriate, and sincerely hope the current version is easier to follow. We will also be happy to edit the paper further based on any further helpful comments from the reviewers. The specific update corresponds to each clarity issue you raised as follows.
>
> >Examples of lack of rigour: the use of the concept of "ideal case" mentioned at the bottom of page 2 to support the idea that Neural FQI converges to optimal Q-function.
>
> **A:** Thanks for pointing out this issue. We have provided more specific explanations, respectively, instead of using ``the ideal case''. This is a commonly used assumption about the richness of function class in statistical learning theory. In Neural FQI, we explain as $\\{Q_\theta: \theta \in  \Theta\\}$ is sufficiently large such that it contains $\mathcal{T}^{opt} Q_{\theta^*}^{k}$, while in Neural FZI and proposition 3, we explain as $\\{Z_\theta: \theta \in  \Theta\\}$ is sufficiently large such that it contains $\{Y_i\}_{i=1}^n$. We also revised it accordingly to the updated version.
>
> >Examples of lack of rigour: a mismatch of assumption between the statement of proposition 3 and its proof ...
>
> **A:** We would like to clarify that $\mathbb{E}\left[Y_i\right]$ can be representable is a corollary if $\\{Y_i\\}$ is representable as the $\mathbb{E}\left[Y_i\right]$ is (the limit of ) the  first component of $\{Y_i\}$'s pdf via our decomposition.
>
>
> >Example of contradiction: Proposition 3 is contradicting itself in equation (6). The left equation claims that the minimizer over converges to the k+1 iterate of the Q function in FQI while on the right claiming that it converges to the actual Q-value function.
>
> **A:** We apologize for this typo, where the action Q-value function should be replaced with the $k+1$ iterate of the Q function, which is thus consistent with the left equation. We have fixed the typo in the revised paper, where, specially, the correct form of the right equation  is updated as:
>
> $$\int_{-\infty}^{+\infty}\left|F_{q_\theta}(x)-F_{\delta_{ \mathcal{T}^{\text{opt}} Q^k_{\theta^*}(s, a)}}(x)\right| d x =o(\Delta)$$
>
>
> >Q: Typos and inclarity include: ...
>
> $A:$ Thank you for noting these typos and we have fixed them appropriately. For example, we acknowledge that p.4 $\Delta_E^i$ represents the interval of $\mathbb{E}[Y_i]$ instead of the current return distribution and added a notation warning about the low case random variables. We also gave a more accurate definition of $\widehat{\mu}$.
>
> Please refer to the updated paper for more details. Please let us know if there are any other typos you had in mind for correction.
>
> We thank the reviewer once again for the time and effort in reviewing our work! As suggested, in the rebuttal period, we have finished the throughout notation revisions to allow an informed evaluation. We would greatly appreciate it if the reviewer could check our responses and are more than happy to provide further clarification if you have any additional technical concerns about our paper.

---

### Author Response · Authors · 2023-11-22
**General Author Response**

We thank all the reviewers for their constructive comments and feedback on our paper. **We have revised the clarity issues throughout the paper and updated our paper accordingly with the modifications in red**. We also provide the original version of our paper in the supplementary for reference.

We believe our paper has significant contributions, mainly including (1) proposing the return density decomposition technique based on the histogram function estimator (2) interpreting distributional RL as the regularization term in the Neural FZI framework and (3) connecting the regularization term in distributional RL with MaxEnt RL (4) derived a practical algorithm that interpolates between expectation and distributional RL.

Our work answered a very important question in the distributional RL community: **how to understand the benefits of distributional RL over classical RL?** by starting from CDRL. We hope our analysis method can also inspire more researchers in the community. We would greatly appreciate it if the reviewers could check our responses and let us know whether they address the raised concerns. We are more than happy to provide further clarification if you have any additional concerns.

---

### Meta-Review · Area_Chair_2ocn · 2023-12-09

**Metareview:**

This paper studies the question of when distributional reinforcement learning outperforms traditional reinforcement learning from a theoretical perspective. Their results suggest an interpretation of distributional reinforcement learning as adding "risk-sensitive entropy regularization", which serves to encourage exploration in the MDP. The authors also perform extensive experiments to evaluate their theoretical claims.

The reviewers generally agreed that while the direction studied by the paper is important, and the results are potentially of interest, the paper is poorly written and needs to be significantly clarified. The reviewers generally found that terms were not precisely defined, contradictions between different parts of the paper, or proofs that were not rigorous, all impeding understanding of the technical contents of the paper. There was also some concern about the significance of the results. While the authors have worked towards improving clarity, significant concerns remain.

**Justification For Why Not Higher Score:**

The clarity of the exposition and lack of mathematical precision and rigor inhibits understanding of the main results of the paper.

**Justification For Why Not Lower Score:**

N/A

---

### Decision · Program_Chairs · 2024-01-16

Reject